# Biogenic SOA sensitivity to organic aerosol simulation schemes in climate projections

Arineh Cholakian[1,2,*], Matthias Beekmann[1], Isabelle Coll[1], Giancarlo Ciarelli[1,**], and Augustin Colette[2]

[1]LISA, UMR CNRS 7583, Université Paris Est Créteil et Université de Paris, Institut Pierre Simon Laplace (IPSL), Créteil, France
[2]Institut National de l'Environnement Industriel et des Risques, Parc Technologique ALATA, Verneuil-en-Halatte, France
[*]Now at EPOC, UMR 5805, Université de Bordeaux, Pessac, France
[**]Now at Department of Chemical Engineering, Carnegie Mellon University, Pittsburgh, USA

**Correspondence:** Arineh Cholakian (arineh.cholakian@lisa.u-pec.fr)

**Abstract.** Organic aerosol (OA) can have important impacts on air quality and human health because of its large contribution to atmospheric fine aerosol, and its chemical composition, including many toxic compounds. Simulation of this type of aerosol is difficult since there are many unknowns in its nature, and mechanism and processes involved in its formation. These uncertainties become even more important in the context of a changing climate, because different mechanisms, and their representation in atmospheric models, imply different sensitivities to changes in climate variables. In this work, the effects caused by using different schemes to simulate OA are explored. Three schemes are used in this work: 1) a molecular scheme, 2) a standard volatility basis set (VBS) scheme with anthropogenic aging and 3) a modified VBS scheme containing functionalization, fragmentation and formation of non-volatile SOA for all semi-volatile organic compounds (SVOCs). 5 years of historic and 5 years of future simulations were performed using the RCP8.5 climatic scenario. The years were chosen in a way to maximize the differences between future and historic simulations. The study focuses on BSOA since the contribution of this fraction of BSOA among OA is major in both historic and future scenarios (40 to 78% for different schemes in historic simulations). Simulated OA and BSOA concentrations with different schemes are different, the molecular scheme showing the highest concentrations among the three schemes. The comparisons show that for the European area, the modified VBS scheme shows the highest relative change between future and historic simulations, while the molecular scheme shows the lowest (a factor of two lower). These changes are largest over the summer period for biogenic SOA (BSOA) because the higher temperatures increase terpene and isoprene emissions, the major precursors of BSOA. This increase is partially off-set by a temperature induced shift of SVOCs to gas phase. This shift is indeed scheme dependent, and it is shown that it is the least pronounced for the modified VBS scheme including a full suite of aerosol aging processes, comprising also formation of non-volatile aerosol. For the Mediterranean Sea, without BVOC emissions, the OA changes are less pronounced and, at least on an annual average, more similar between different schemes. Our results warrant further developments in organic aerosol schemes used for air quality modelling to reduce their uncertainty, including sensitivity to climate variables (temperature).

# 1 Introduction

Organic aerosol (OA) is an important fraction of fine particulate matter (PM) concentrations. Its production results from both primary emissions of organic aerosols, as well as secondary formation from semi-volatile or polar precursor gases in the atmosphere. The mechanisms and pathways of secondary organic aerosol (SOA) formation are in general highly uncertain (Hallquist et al., 2009, Tsimpidi et al., 2017). Yet, the importance of the concentrations of OA in the atmosphere (Jimenez et al., 2009) and their adverse effects on human health (Mauderly and Chow, 2008, Lelieveld et al. 2015) make them an important subject to study.

Considering that modelling OA already contains important uncertainties, the uncertainties become even more important for future climate scenarios which account for climate change. These future scenarios present an important number of uncertainties, both due to climate related parameters, but also due to the description of how they act on specific processes. As an example, biogenic volatile organic compound (BVOC) emissions, which are the main precursors of biogenic SOA (BSOA), can be affected by e.g. temperature and land use changes, $CO_2$ inhibition (Heald et al., 2008a) among other factors. Many studies have addressed the effects of these parameters on the BVOC emissions, and a high variability was found in BVOC emissions depending on the factors that were considered in each study. For example, Heald et al. (2009) explored the effects of land use change and $CO_2$ inhibition on the emission of BVOCs and they found a 130% of isoprene emission increase in 2100 compared to 2000, while Pacifico et al. (2012) and Hantson et al. (2017) show 70% and 41% increase for isoprene for the same years with different parameters. Langner et al. (2012) compares four different models for the European region reporting an isoprene increase in the range of 21%-26%. Cholakian et al. (2019), found an increase of 52% for isoprene for the period of 2031-2100 compared to 1976-2005 because of only temperature change for Europe, amounting to a 12% increase in BSOA concentrations.

In addition, for the formation of anthropogenic SOA (ASOA), future urbanization, anthropogenic emission and wood burning emission changes can be mentioned as possible factors. Each one of these parameters represents an uncertainty, which, when coupled with the inherent uncertainty in the simulation of OA, can present important sources of error.

It is mainly to assess the future evolution of tropospheric ozone that BVOC emissions have been quantified at global scale in chemistry-climate projections (Arneth et al., 2010). Their importance for organic aerosol chemistry has also been considered in global and regional scale atmospheric models (Maria et al., 2004; Tsigaridis et al., 2007; Heald et al., 2008b), but to a lesser degree. Several different types of OA simulation schemes can be used in chemistry-transport models (CTMs). Odum (1997) suggested a two-product scheme, where he calculated yields of production of OA from VOCs from laboratory data. He concluded that two virtual semi-volatile organic compounds were sufficient to represent the formation of OA. Following the partitioning theory of Pankow (Pankow, 1994), these species are distributed between the aerosol and gas phases. Pun and Seigneur, (2007) suggested a molecular single-step oxidation scheme for the formation of SOA, based on the Odum scheme. Another approach is the volatility basis set (VBS) scheme, which includes different volatility bins and aging of semi-volatile species lowering their volatility (Donahue et al., 2006; Robinson et al., 2007). This scheme presents two major versions: 1-dimentional (1D) and 2-dimentional (2D) VBS. 1D-VBS distributes semi-volatile organic compounds (SVOCs) into different bins with regards to their volatility (Robinson et al., 2007). A 2D-VBS scheme, takes into account the oxygen to carbon

(O/C) ratio as well as the volatility (Donahue et al., 2011; Donahue et al., 2012). While 1D-VBS has been tested extensively in different CTMs (i.e Lane et al., 2008; Hodzic and Jimenez, 2011; Zhang et al., 2013; Cholakian et al., 2018), the use of 2D-VBS is less frequent because of its even more challenging numerical needs. Other variations of the 1D-VBS have been also used for observation-simulation comparisons, each one adding some variables to the basic VBS scheme or building upon

its framework. For example, Shrivastava et al. (2015) adds fragmentation and formation of nonvolatile SOA mechanisms to the basic 1D-VBS scheme. This scheme was implemented into the CHIMERE CTM and tested for the Mediterranean region with good results in terms of concentration (correlation of 0.55 and a bias of -0.68$\mu g$ $m^{-3}$ for the summer period of 2013), fossil/non-fossil distribution and oxidation level of OA (Cholakian et al., 2018). Besides, Lannuque et al. (2018) provide a new parameterization for the VBS scheme by using a box model based on the GECKO-A modelling tool, which was afterwards

implemented in CHIMERE and tested for the European continent, showing a good correspondence between modeled and measured OA (Lannuque et al. 2019, in prep).

In addition, the sensitivity of OA schemes to thermodynamic parameters could show large differences due to different processes considered or due to the differences in the parameterization. The formation and partitioning of particulate OA can show various degrees of dependency to temperature in different OA schemes. Therefore, the sensitivity of organic aerosol to

climate change, affecting these thermodynamic parameters (mainly temperature), also depends on the OA scheme used. To our knowledge, this issue has not yet been addressed in a dedicated work. In most future scenarios, a two-product scheme is used for the simulation of SOA. However, other schemes, such as different variations of the VBS scheme, could better represent the more complex characteristics of SOA, such as, for example, its oxidation state.

Differences induced by different schemes are also expected to vary regionally, depending on the concentration ranges en-

countered and ranges and changes in meteorological parameters. In this study, we focus on the European continent and the Mediterranean basin. The Mediterranean basin, is one of the most sensitive regions to climate change, which makes it important and at the same time interesting to study. However, not much focus has been given to the Mediterranean in the literature, especially for the western side of this basin (Giorgi, 2006). For this reason, the ChArMEx project was put into place, in order to study the current chemical characteristics of the atmosphere of the Mediterranean region and its changes in future scenarios.

In this study, future OA concentrations under a climate change scenario will be quantified using different OA schemes. Three OA simulation schemes are compared, namely (i) a two-product scheme, (ii) a VBS scheme with anthropogenic aging and (iii) a modified VBS scheme including fragmentation and nonvolatile SOA formation. A representative concentration pathway climatic scenario (RCP) has been used. RCP8.5 has been chosen in order to maximize future changes and to get a clear climate change related signal in our study.

The paper is organized as follows: Section 2 explains the modeling framework for this work. An evaluation of the three schemes against measurements is provided in section 3, while section 4 presents results for the different scenarios. Conclusions are presented in section 5.

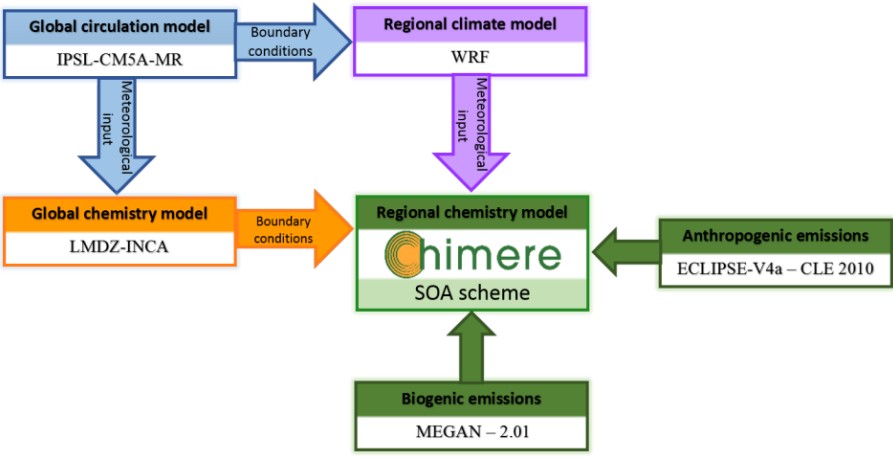

**Figure 1.** Simulation chain used for this study: the focus of this work is the SOA scheme inside the regional chemistry transport model.

## 2 Simulations

The modelling framework in this study utilizes a chain of models, covering the different compartments of the atmosphere, a global circulation model and a global chemistry transport model providing meteorological and chemical conditions of the atmosphere respectively (figure 1). In order to down-scale the output provided by the global models a regional climate model

and a regional chemistry transport model are used (figure 1). Global circulation data is provided by IPSL-CM5A-MR (Taylor et al., 2012 ; Dufresne et al., 2013 ; Young et al., 2013), while the LMDZ-INCA (Hauglustaine et al., 2014) global chemistry transport model, using simulations from global circulation model as meteorological input, provides boundary conditions for the regional chemistry transport model (CTM). The boundary conditions include inputs for organic carbon as well. The global circulation model also provides boundary conditions for the regional climate model, WRF (Weather Research and forecasting,

Wang et al., 2015), which, in return provides meteorological input fields for the regional CTM, CHIMERE (Menut et al., 2013). The WRF simulations were prepared for the EURO-CORDEX project (Jacob et al., 2014) and use representative concentration pathways (RCPs, Meinshausen et al., 2011 ; van Vuuren et al., 2011) for future simulations. The EURO-CORDEX climatic runs were performed for the period of 1976—2005 for historic simulations and 2031—2100 for future scenarios, for RCP2.6, RCP4.5 and RCP8.5. A detailed analysis of these runs is provided in Vautard et al. (2014) and Jacob et al. (2014). In this

work, the RCP8.5 runs are used for a selection of years (section 2.3). Anthropogenic emissions (base year 2010) are taken from the ECLIPSEv4a inventory (Amann et al., 2013; Klimont et al., 2013; Klimont et al., 2017), and the biogenic emissions are calculated with the MEGAN model (Model of Emissions of Gases and Aerosols from Nature, Guenther et al., 2006). The coupling of all these models with the CHIMERE model is done in an off-line fashion, except for MEGAN which is directly coupled with CHIMERE. Since the focus of this article is on the SOA scheme changes in the regional CTM, only this model

will be discussed in further detail. More information on the modeling framework in the current study is provided in Colette et al., 2013; 2015.

## 2.1 CHIMERE chemistry transport model

The CHIMERE chemistry transport model has been widely used in different parts of the world (Carvalho et al., 2010; Hodzic and Jimenez, 2011), especially in Europe (Zhang et al., 2013 ; Petetin et al., 2014; Colette et al., 2015; Menut et al., 2015 ; Rea et al., 2015), for both forecasting and analysis purposes. It provides a wide range of capabilities; if input information such as anthropogenic/biogenic emissions, meteorological conditions are given, it can simulate an exhaustive list of atmospheric components. Different chemistry schemes are available in the model, in the case of our simulations, the MELCHIOR2 scheme (Derognat et al., 2003) is used, containing around 120 reactions. A sectional logarithmic aerosol size distribution of 10 bins is used with a range of $40nm$ to $40\mu m$. The aerosol module in CHIMERE includes different chemical and physical processes such as gas/particle partitioning, coagulation, nucleation, condensation, as well as dry and wet deposition. The chemical speciation contains EC (Elemental Carbon), sulfate, nitrate, ammonium, SOA/SVOC species, dust, salt and PPM (primary particulate matter other than ones mentioned above). More information on the SOA scheme will be provided in the next section. The simulation domain covers the whole Europe with a resolution of 0.44°, the domain used in all the simulations are all the same (the domain approximately covers 30-70°N and 40W-60E).

## 2.2 OA schemes used for the simulations

The CHIMERE model has three SOA simulation schemes with different levels of complexity, all based on a molecular single-step oxidation scheme. In our base simulations, the medium complexity scheme is used (Bessagnet et al., 2008). In this scheme, lumped volatile organic compounds (VOCs) can react and form classes of organics with reduced volatility, i.e. SVOCs. Once formed, the model distributes these species between the gaseous and particulate phases according to the mixing theory of Pankow (Pankow, 1987). The yields for the formation of SOA are taken from Odum et al. (1997), Griffin et al. (1999) and Pun and Seigneur (2007). This scheme is referred to as the SOA2p scheme here after. A large database of historic and future simulations exists for this scheme, for three RCPs (RCP2.6, RCP4.5 and RCP8.5), each containing 70 years of simulation (2031 – 2100) and 30 years (1976 – 2005) of historic simulations. These scenarios are discussed and compared in Colette et al. (2013a), Lemaire et al. (2016) and Cholakian et al. (2019) in more detail.

The VBS approach was developed as a general framework to account for the semi-volatile character of organic matter and to allow for changes in volatility over time. In VBS schemes, the SVOCs are partitioned into bins according to their saturation concentrations. Aging processes included by transferring species from one volatility bin to another (Robinson et al., 2006). This scheme was implemented into CHIMERE and tested for Mexico City (Hodzic and Jimenez, 2011) and the Paris region (Zhang et al., 2013). Nine volatility bins with saturation concentrations in the range of 0.01 to $10^6 \mu g$ $m^{-3}$ are taken into account and the emissions of SVOC and IVOC (Intermediate Volatility Organic Compounds) are distributed into these bins using the aggregation proposed by Robinson et al. (2007). Four volatility bins are used for ASOA and BSOA ranging from 1 to $1000\mu g$ $m^{-3}$. Since the aging processes of biogenic SOA were reported to overestimate the BSOA concentrations in

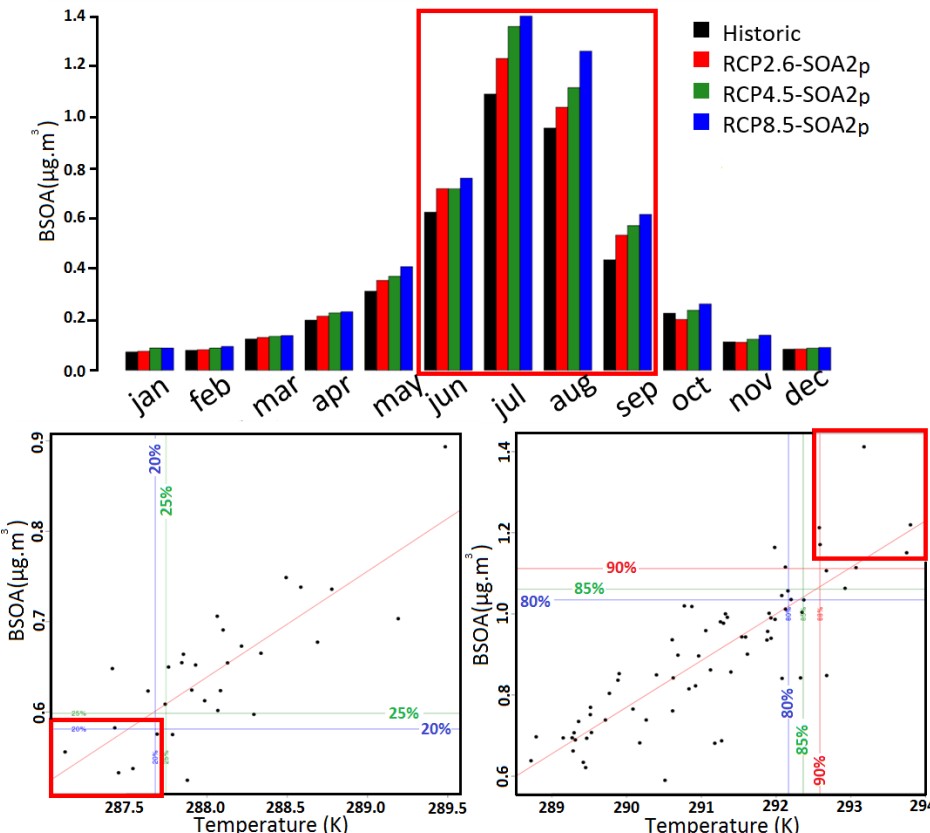

**Figure 2.** Monthly BSOA concentrations in different RCP scenarios, averaged over 70 years of simulations for future scenarios and 30 years for historic simulations (upper panel). BSOA for the June to September period highlighted in the upper panel was plotted against temperature and years with the lowest temperature and lowest BSOA concentrations for historic simulations are shown in the lower panel on the left side. Those with the highest temperature and BSOA concentrations for future scenarios are shown on the right side.

CTM runs for North America (Robinson et al., 2007; Lane et al., 2008) and the Mediterranean Sea (Cholakian et al., 2018), these processes are not taken into account in this work. Gaseous-particulate partitioning is treated following Raoult's law and depends on total organic aerosol concentrations.

Since the standard VBS scheme does not include fragmentation processes (when molecules break into smaller and more volatile molecules in the atmosphere) explicitly and the formation of non-volatile SOA (when SOA, after their formation, become irreversibly non-volatile and therefore cannot be oxidized further), these processes were added to the basic VBS scheme following Shrivastava et al. (2011; 2013; 2015). Another change made to the VBS scheme was to include an interpolation between high-$NO_x$ and low-$NO_x$ regimes (Carlton et al., 2009). Reaction rates for the common species (and generations) do not change between these two schemes, reaction rates for new species and generations is taken from Shrivastava et al. (2013). The aging processes are all turned on in the modified VBS scheme, two more oxidation generations are added to POAs. BSOA

oxidation generations are kept the same (one generation of oxidation). The formation of non-volatile SOA is added to all the SOA oxidized species (excluding POA), forming a nonvolatile SOA which cannot return to the gaseous phase. The same fragmentation fractions reported by Shrivastava et al. (2015) are used without any change.

Both the standard VBS without biogenic aging (referred to as SOAvbs scheme here-after) and the modified VBS including fragmentation and formation of non-volatile aerosol (referred to as SOAmod scheme here-after) schemes are presented in more detail and compared to experimental data in the western Mediterranean area in Cholakian et al. (2018). In the aforementioned work, it was concluded that these two schemes can reproduce the levels of concentration of organic aerosols in the Mediterranean basin successfully in regards to concentration of OA, while oxidation state and fossil/non-fossil repartition is better represented in SOAmod.

## 2.3 Choice of years

The SOAvbs and the SOAmod schemes are both numerically very resource-consuming, therefore, only 10 years of simulations for each scheme were performed. In order to choose the appropriate years for the simulation, existing long-period sets of simulations were used, containing 30 years of historic simulations (1976-2005) and 70 years of future scenarios (2031-2100). We address results for BSOA, as it makes the major contribution to OA during summer (between 40 and 78% for different schemes in the historic scenario, according to our simulation results for the historic period with differences schemes, not shown in figures).

The simulations were performed using the previous version of CHIMERE (chimere-2013b, Menut et al 2013), the SOA2p scheme and the RCP8.5 scenario. This dataset was used to choose five years of simulations in the historical and future periods each, with the aim to maximize both the temperature and SOA differences between historic and future scenarios. Figure 2-a shows the monthly average of BSOA concentrations in different RCP scenarios, showing that the production of BSOA reaches its maximum in the period of 4 months of June, July, August and September. During these months BSOA is the major SOA and OA component over Europe as also discussed in Cholakian et al. (2019).

Figure 2-a also displays that the differences of historic and future simulations reaches its maximum for RCP8.5 simulations. The concentration of BSOA and the temperature both in historic and RCP8.5 simulations show a strong positive correlation as seen in figure 2-b and 2-c, each point representing the average of the four months mentioned previously for one year.

For historic simulations, the years representing the lowest temperature and BSOA concentrations are used, which correspond to years 1980, 1981, 1984, 1985 and 1986, while for future scenarios the years with the highest temperature and BSOA concentrations are used corresponding to years 2087, 2092, 2093, 2095 and 2098.

## 3 Scheme validation

The three schemes show high variability when simulating the concentration and characteristics of OA, therefore, we performed an evaluation to investigate their performances. The schemes are compared to observations for the year 2013 during which an abundance of observational data is available. A year-long simulation for the year 2013 was performed for each of the schemes.

The inputs used in these simulations are the same: anthropogenic emissions are taken from EMEP (European Monitoring and Evaluation Programme, http://www.ceip.at), meteorological fields are generated using the ECMWF (European Centre for Medium-Range Weather Forecasts) input data (Berrisford et al., 2011), biogenic emissions are provided by MEGAN (Guenther et al., 2006) and boundary and initial conditions are taken from LMDZ-INCA (Hauglustaine et al., 2014).

The observations are mostly accessed from the EBAS database (http://ebas.nilu.no/, Tørseth et al., 2012). The used measurements are mostly $PM_{2.5}$, in some cases $PM_1$. For each type of measurement ($PM_{2.5}$ or $PM_1$) the corresponding fraction from the simulations has been used. In some cases, data was provided by the lead investigator for a specific station, and the measurements for the two stations of Corsica and Mallorca have been added using the ChArMEx (http://mistrals.sedoo.fr/ChArMEx/) campaign measurements. In total, 32 stations are compared to simulations. Bear in mind that for some of these stations the available data covers a shorter period than one year, or they present weekly measurements rather than daily observations. The list of stations with information about each station and the type of measurement is provided in appendix A.

Results of these comparisons are shown in figure 3 and the statistic information is shown in table 1. Regarding the concentration of OA, the modified VBS scheme shows a stronger bias ($-0.64 \mu g\ m^{-3}$ compared to $0.42 \mu g\ m^{-3}$ and $0.1 \mu g\ m^{-3}$ for SOAvbs and SOA2p respectively) for the summer period. All compared schemes underestimate the winter period ($-1.45$, $-1.67$ and $-1.63 \mu g\ m^{-3}$ for SOAmod, SOAvbs and SOA2p respectively). The annual biases for the three schemes are $-0.91$, $-0.4$ and $-0.65 \mu g\ m^{-3}$ for SOAmod, SOAvbs and SOA2p respectively. The correlation of determination between observed and simulated OA concentrations for different schemes are the highest for the SOAmod, and lowest for the SOAvbs in most seasons; it should also be noted that the difference between the correlations seen for each scheme are rather small (difference of below 0.05) . The Taylor diagram in figure 3 shows the comparisons of different stations to simulations for each scheme (black for SOA2p, green for SOAmod and red for SOAvbs). The Taylor diagram summarizes several statistic information in one plot: The correlation coefficient, root-mean-square (RMS) difference between observations and simulations and the standard deviation ratio can be seen. More information about the construct of this diagram is given in Taylor, 2001. As seen in this diagram, there is a high variability in the simulation of different stations, some stations are better represented by the model than the others. This might be because of the geographical placement (altitude, types of emissions in said location, etc.) of stations or because of the nature of the station (urban, rural, etc.).

The three schemes perform reasonably well according to the criteria introduced by Boylan and Russell (2006), with the values for all the schemes falling into in zone 1 for both mean fractional bias (MFB) and mean fractional error (MFE). The goal for these two metrics according to the aforementioned reference is less than or equal to ±30% and +50% and the criteria is less than or equal to ±60% and +75% respectively. The MFB values for the three schemes are -19.7%, 16.5% and 26.9%, while MFE shows 47.9%, 51.1% and 47.2% for SOA2p, SOAvbs and SOAvbs respectively. Thus, performance goals are met nearly for all schemes, with a slight exceedance for MFE and the SOAvbs scheme, still meeting the criterium. Each one of the schemes performs better for a specific period; modified VBS in summer, CHIMERE standard scheme during winter, and the standard VBS scheme showing average performance during the whole year. Looking at table 1, it is seen that for example for the summer period, the SOAmod scheme shows the highest correlation of determination, while SOA2p shows the lowest bias for this season. For spring SOA2p shows the highest $R^2$, while SOAvbs shows the lowest bias. For winter and autumn,

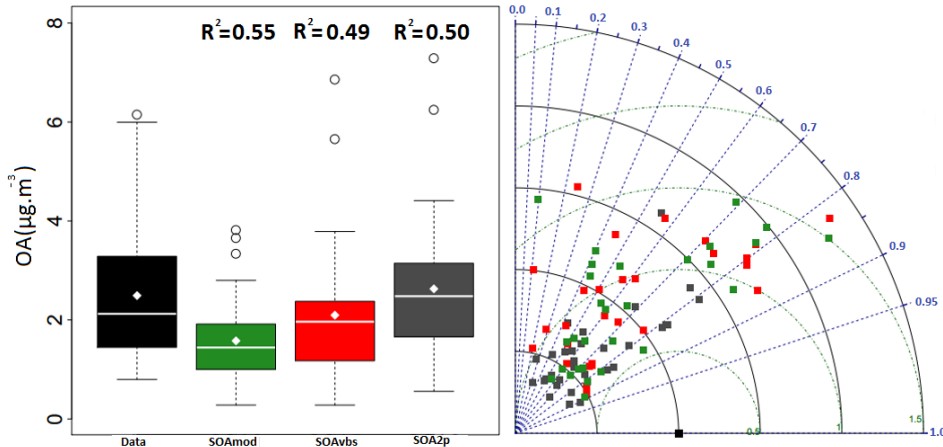

**Figure 3.** Observation-simulation comparisons for different schemes: The boxplots are presented for the data and the schemes for the annual averages, the correlation of determination for each scheme is shown above each boxplot. The point on the boxplot shows the average value. Each point in the Taylor diagram shows one station while each color shows one scheme (black: SOA2p, red: SOAvbs and green: SOAmod). Table 1 shows statistic information for observed data and the three studied schemes.

the performance of the schemes is quite similar. Annually, SOAvbs presents a similar correlation to SOA2p while showing the lowest bias in general. The types of stations have not been filtered in the current study, therefore, all stations, including urban, semi-rural or rural have been included for the comparisons. This could be responsible for part of the observed negative bias. As a conclusion, all three schemes correspond to the performance goals and/or criteria of Boylan and Russell (2006), albeit they show important, and spatially and seasonally dependent, differences with observations. Thus, the three schemes will be retained for the following analysis with equal confidence.

## 4 Analysis of the simulations

The analysis of the simulations will be presented in the next two sub-sections. First, the changes in BVOC emissions are discussed. Subsequently, the results for the European continent regarding concentration, origins and oxidation state will be presented. An analysis of these parameters will be performed for the Mediterranean sub-domain including their origins and the oxidation state. Finally, a general comparison of the spatial distribution will be performed for different schemes. It is important to keep in mind that from this section on, whenever BSOA concentrations are discussed the $PM_1$ fraction of this species has been used.

### 4.1 Changes in biogenic emissions

The changes in biogenic emissions are important in the context of this work, since they are highly dependent to temperature changes. For the simulations presented in this work, the biogenic emissions do not change between different schemes, however

|  |  | Average | Bias | Standard deviation | RMSE | $R^2$ |
|---|---|---|---|---|---|---|
|  |  |  |  | μg.m$^{-3}$ |  |  |
| Annual | Data | 2.49 |  | 0.36 |  |  |
|  | SOA2p | 1.84 | -0.64 | 1.40 | 2.21 | 0.49 |
|  | SOAvbs | 2.1 | -0.4 | 1.62 | 2.44 | 0.50 |
|  | SOAmod | 1.58 | -0.91 | 1.05 | 2.11 | 0.55 |
| DJF | Data | 2.55 |  | 1.79 |  |  |
|  | SOA2p | 0.91 | -1.63 | 0.50 | 2.47 | 0.51 |
|  | SOAvbs | 0.89 | -1.67 | 0.65 | 2.63 | 0.51 |
|  | SOAmod | 1.11 | -1.45 | 0.75 | 2.55 | 0.56 |
| MAM | Data | 2.11 |  | 1.33 |  |  |
|  | SOA2p | 1.88 | -0.23 | 1.3 | 1.81 | 0.56 |
|  | SOAvbs | 2.23 | 0.11 | 1.42 | 1.98 | 0.50 |
|  | SOAmod | 1.4 | -0.71 | 0.84 | 1.64 | 0.49 |
| JJA | Data | 2.48 |  | 1.12 |  |  |
|  | SOA2p | 2.56 | 0.1 | 1.71 | 2.06 | 0.51 |
|  | SOAvbs | 2.9 | 0.42 | 1.91 | 2.31 | 0.54 |
|  | SOAmod | 1.83 | -0.65 | 1.17 | 1.69 | 0.58 |
| SON | Data | 2 |  | 1.36 |  |  |
|  | SOA2p | 1.16 | -0.84 | 0.81 | 1.81 | 0.47 |
|  | SOAvbs | 1.28 | -0.72 | 0.92 | 1.85 | 0.47 |
|  | SOAmod | 1.3 | -0.7 | 0.81 | 1.76 | 0.45 |

**Table 1.** Statistic information for different schemes in regards to measurements. In this table are provided the average for the data and each scheme, the bias, standard deviation and RMSE (all in $\mu g\ m^{-3}$. $R^2$ shows the correlation of determination for in scheme. Figure 3 shows the taylor diagram and the boxplots for these comparisons.

they change quite a bit between historic and future simulations because of temperature increase in the future. Since the choice of the years was done to maximize future temperature changes, the differences between future and historic simulations are quite remarkable. For the European region, the average historic isoprene emissions are $1.3 \times 10^{11} molecules\ cm^{-2}\ yr^{-1}$ and average historical terpene emissions are $3 \times 10^{10} molecules\ cm^{-2}\ yr^{-1}$. An increase of 88% and 82% for isoprene and terpenes is seen respectively in the future scenarios in response to an average temperature increase of 5.5°C. For the summer period, the biogenic emission increase raises to 93% and 92% for isoprene and terpenes for a temperature increase of 6.4°C (figure 6). The correlation of determination between historic isoprene and terpene emissions is 0.6 and 0.63 while this correlation is 0.65 and 0.57 for the future simulations.

For the Mediterranean Sea, there are no local biogenic emissions included in the model.

## 4.2 European region

### 4.2.1 Changes in BSOA concentration

BSOA concentrations in future scenarios are predicted to increase in all the schemes. However, the intensity of this increase is scheme dependent: while for SOA2p an annually averaged increase of +61% is calculated, this percentage rises to +80% for SOAvbs and +98% for SOAmod for the same period. These changes show that the climate impact on changes of BSOA in the future might be underestimated until now on a relative scale. This is because most of the future simulations performed in order to explore climate impact use a two-product or a molecular single step scheme for the simulation of SOA. However, our study shows that using a VBS based scheme increases the climate induced effect on the change in BSOA concentration in the future. Reasons for this behavior will be discussed in section 5. However, we would like to emphasize that changes are maximized by the choice of the RCP8.5 scenario and the years chosen for the simulations in this work. Also, it should be noted that there are important differences in absolute concentrations between different schemes (see above).

There is a strong seasonality for the BSOA production. The seasonal changes for BSOA are seen in figure 4-a1, 4-b1 and 4-c1 for historic simulations, the absolute difference between future and historic simulations, and their relative changes respectively. Summer shows the maximum relative increase (+113%, +155% and +262% for SOA2p, SOAvbs and SOAmod respectively) and winter the lowest one in all schemes (+31.1%, +26.2% and +20.5% for SOA2p, SOAvbs and SOAmod respectively). For autumn and spring SOA2p and SOAmod show similar and intermediate changes while SOAvbs shows higher relative differences (+59.6%/+40.3%, +79.9%/+60.0% and +57.3%/+50.0% for SOA2p, SOAmod and SOAvbs respectively for autumn/spring).

For monthly results, as seen in figure 4-a2, 4-b2 and 4-c2 there is an increase (both in relative and absolute values) in almost all months for all schemes during the year, but the intensity of this increase changes for different months. In July, when the BSOA concentration reaches its maximum, the percentage of change in the future is high as well (+115%, +151% and +243% for SOA2p, SOAvbs and SOAmod respectively). Highest relative changes occur for August for all schemes (+111%, +165% and +356% for SOA2p, SOAvbs and SOAmod respectively) resulting from the seasonal profile of BVOC emissions.

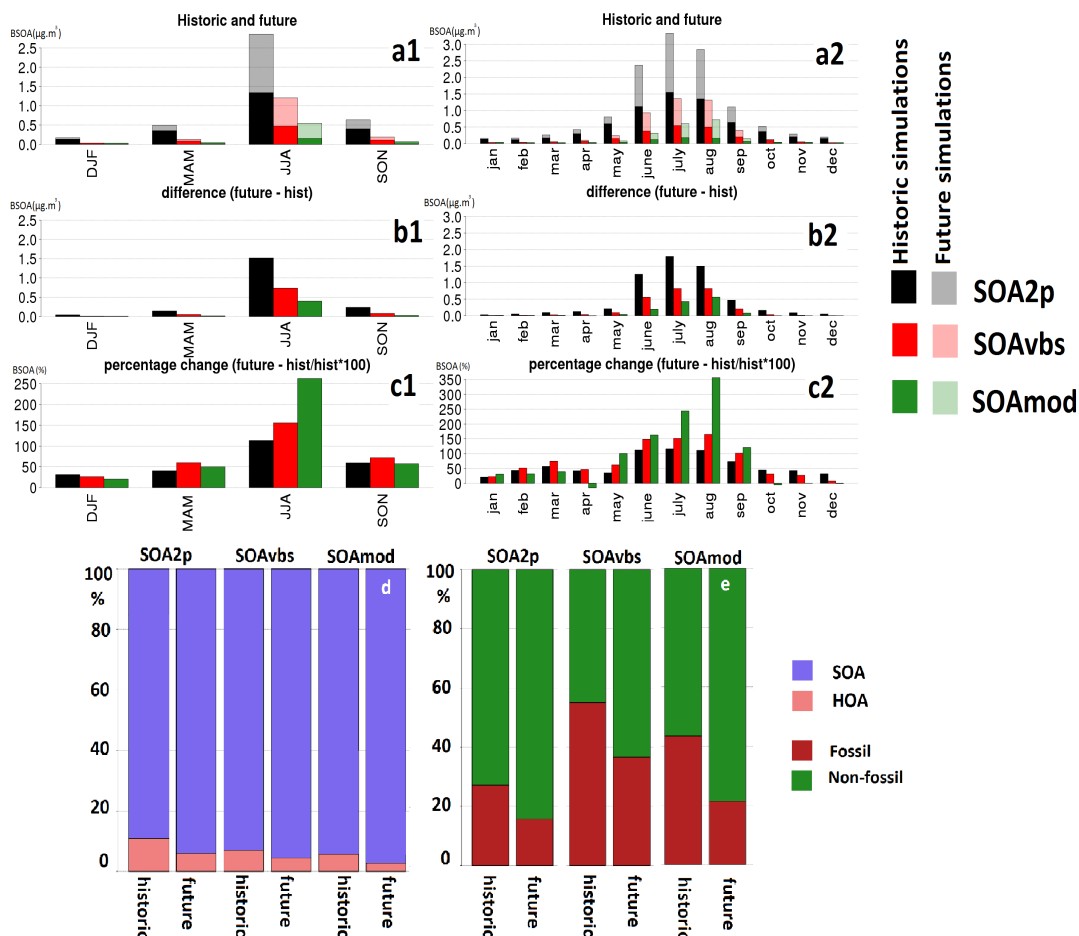

**Figure 4.** Seasonal and monthly averaged concentrations of BSOA for 5 years of simulations and the oxidation state and origins of OA for the European sub-domain. a1, a2 → historic simulations. b1, b2 → Absolute changes in future scenarios compared to historic simulations (future – historic), c1, c2 → Relative changes in future scenarios. d, e → Oxidation state and origins respectively. Lighter colors show future scenarios and darker colors the historic simulations.

For SOAmod, a decreased change is seen for some months in the future scenarios (-11%, -1.6% and -0.45% for April, October and November respectively).

### 4.2.2   Changes in the origins of OA

Figure 4-d shows a simplified distribution for the OA in different schemes: SOA and HOA (hydrocarbon-like organic aerosol) presenting the freshly emitted primary OA. Figure 4-d shows that the predicted distribution between HOA and SOA is different for the three schemes. SOA2p indicates a smaller contribution of SOA and a larger one from HOA compared to SOAvbs and SOAmod schemes. This is because POA emissions in SOA2p are considered non-volatile, while they are volatile in VBS

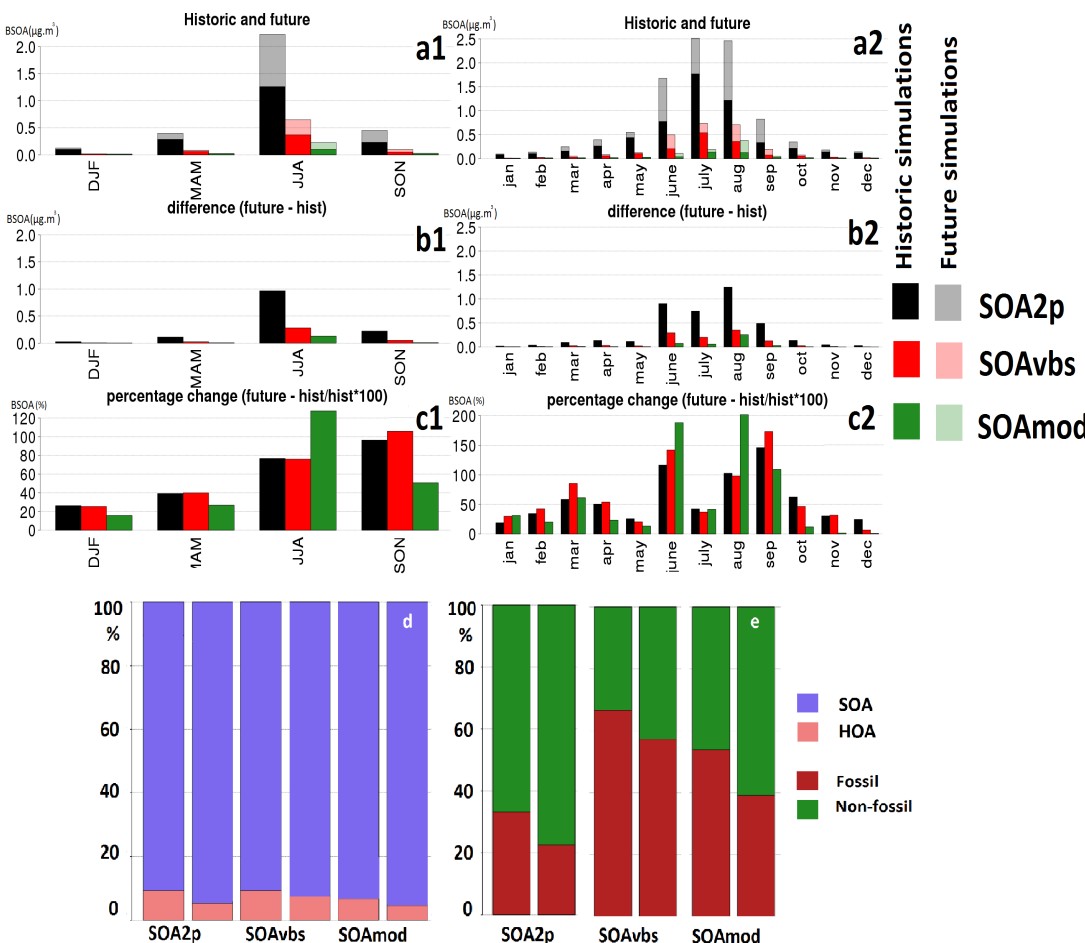

**Figure 5.** Seasonal and monthly averaged concentrations of BSOA for 5 years of simulations and the oxidation state and origins of OA for the Mediterranean sub-domain. a1, a2 → historic simulations. b1, b2 → Absolute changes in future scenarios compared to historic simulations (future – historic), c1, c2 → Relative changes in future scenarios. d, e → Oxidation state and origins respectively. Lighter colors show future scenarios and darker colors the historic simulations.

schemes. The relative contribution of HOA decreases in all schemes in the future scenario, since the anthropogenic emissions are kept constant, and the concentration SOA increases. However, the decrease in the relative contribution of HOA is stronger for the SOAmod scheme, since it shows a higher relative increase in the formation of BSOA in future scenarios.

The schemes behave differently in contribution of different origins in the formation of OA as well, therefore it is interesting to compare this aspect in the tested schemes. Since surrogate species for different sources are present in the outputs, the fossil/non-fossil repartition can be easily calculated. ASOA is considered to be in the fossil fraction (neglecting a small fraction due to bio-fuels) and BSOA in the non-fossil fraction. For carbonaceous aerosol, residential/domestic uses are considered as non-fossil as they are mostly related to wood burning (Sasser et al., 2012). When comparing the simulated fossil/non-fossil

fraction, some differences are observed. The SOAvbs scheme predicts more SOA in the fossil fraction mainly because it takes into account aging of anthropogenic SVOCs and not of biogenic SVOCs. On the contrary, the SOAmod scheme takes into account the aging for both biogenic and anthropogenic SVOCs, therefore it simulates more in the non-fossil compartment. All schemes show a relative increase in the contribution of non-fossil sources in the future on an annually averaged basis (10%, 17% and 22% of increase for the non-fossil partition between future and historic simulations for SOA2p, SOAvbs and SOAmod respectively). As already discussed, a strong seasonality is seen for this factor as well. The contribution of non-fossil sources becomes much higher in summer (figure 4-e), when BVOC emissions are largely abundant. The increase in the contribution of non-fossil sources is logical since the anthropogenic emissions of OA precursors are kept the same and the biogenic emissions of these species increase with increasing temperature.

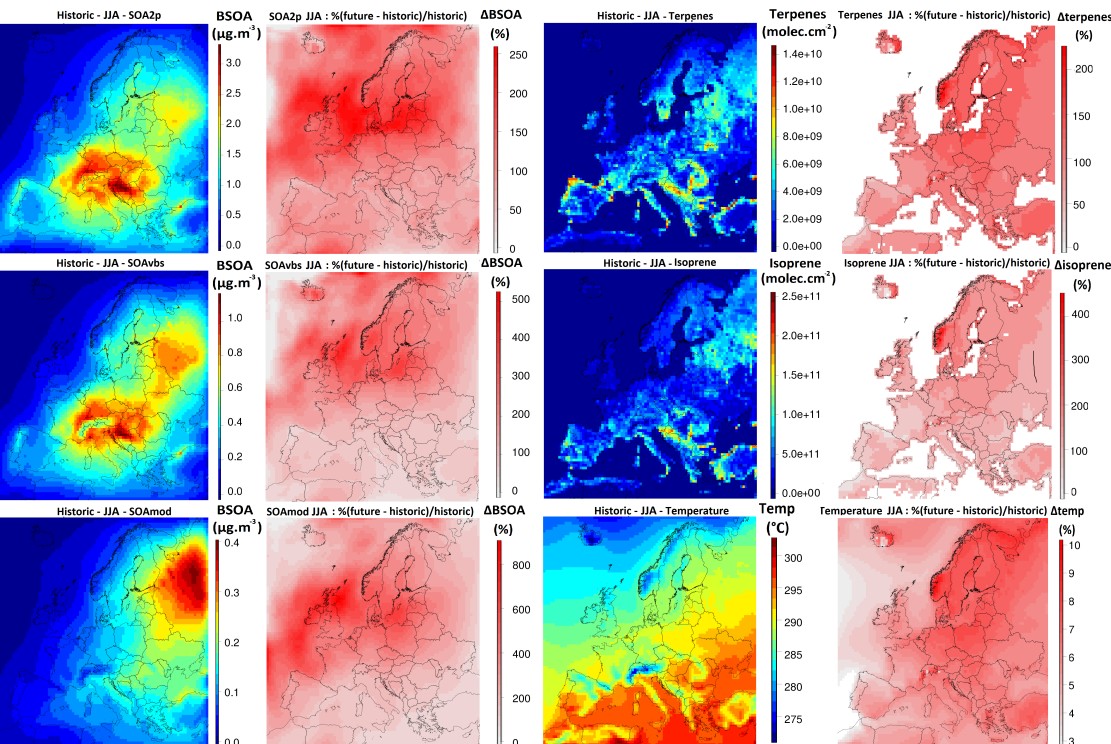

**Figure 6.** Averaged summer changes in concentrations of BSOA in historic (first column, $\mu g\ m^{-3}$) and their future changes (second column, %( future – historic)/historic) for all three scenarios (SOA2p, SOAvbs and SOAmod in first, second and third rows respectively). Third column shows the emissions of mono-terpenes and isoprene ($molecules\ cm^{-2}\ yr^{-1}$, first and second row) and temperature (K, third row) and the changes of each one of these parameters is seen in fourth column (%(future – historic)/historic)). Bear in mind that emissions of BVOCs and the temperature do not change between different schemes, scale for each plot is different and all the figures are for the summer period.

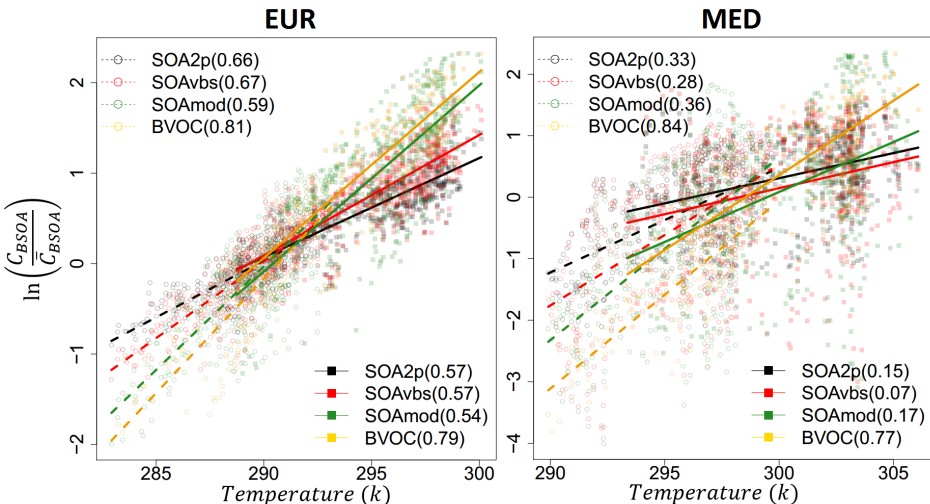

**Figure 7.** Normalized (divided by the average) concentrations of BSOA versus temperature for the summer period (SOA2p in black, SOAvbs in red and SOAmod in green). Each point represents one day of simulation. Empty circles/dashed lines show the historic period, while filled rectangles/filled lines show the future scenarios. The points concerning BVOCs have been added as well (in gold). The regression lines are exponential. The correlation coefficients for each of the schemes are reported in the legend. Emissions of BVOCs are kept constant between different schemes. A different scale is used for each sub-domain to facilitate the comprehension of the panel.

## 4.3 Mediterranean region

While the differences between the schemes for the European area are important to explore in future scenarios, we also focus on the Mediterranean region because of several reasons: high sensitivity to climate change, high burden of OA (and PM in general, Lin et al., 2012; Lin et al., 2014) and also high temperatures in the area. Because of these reasons, we perform
a similar analysis as in the previous section. As explained before, a land-sea mask has been used in order to separate the Mediterranean Sea, therefore the analysis explained below regards only the Sea without any land surface cells.

### 4.3.1 Changes in BSOA concentration

There are major differences between the concentrations of different aerosol components over the Mediterranean area compared to continental Europe. For example, the concentrations of salt and dust particles are higher, for the former because of the marine
environment and for the latter because of the North African dust emissions which are transported to the Mediterranean area. On the contrary, the concentrations of nitrate and BSOA are lower than in the continental area; in the case of nitrate particles, because of higher temperatures its formation is less efficient than it is in continental Europe, and for the BSOA because of lack of emission sources over the marine environment. The differences seen for BSOA concentrations in different schemes are presented in figure 5 (panels 5-a1, 5-b1 and 5-c1 for seasonal results and 5-a2, 5-b2 and 5-c2 for monthly results). The
behavior of different schemes in regards to differences between historic and future simulations differs between the Sea and the

continental area. For BSOA relative changes, SOAmod still shows the largest relative change in the summer period compared to historic simulations (76%, 75% and 127% for SOA2p, SOAvbs and SOAmod respectively), but the differences between schemes are less pronounced in the Mediterranean area.

### 4.3.2    Changes in the origins of OA

For all three schemes, the contribution of fossil sources to OA is slightly larger for the Mediterranean sub-domain than for Europe (figure 5-e). The reason for this difference is the fact that there are local fossil OA formation sources in the Mediterranean Sea, i.e. shipping emissions, while OA originating from non-fossil sources are not directly emitted in this area and are transported from outside. While the contribution of non-fossil sources increases in the future scenarios, fossil sources still contribute more in the Mediterranean area compared to the European area, relative to non-fossil sources.

Both for the historic and the future simulations, the HOA/SOA distribution does not change considerably in the Mediterranean area as compared to the European area (figure 5-d).

### 4.4    Spatial distribution of future changes

Figure 6 shows the concentration of BSOA in different schemes (in $\mu g \ m^{-3}$ first column), the percentage of differences between historic and future simulations (second column), concentrations of isoprene and mono-terpenes and temperature for all schemes

in the third column and the changes of these parameters in future scenarios in the fourth column. All the panels in figure 6 show the summer period. The concentration of BSOA in SOA2p simulations is much higher than that of SOAvbs and even more so than that in SOAmod at the lower end. However, the predicted increase for the future is higher for SOAvbs and SOAmod (figure 6, second column), reaching an average of 290% increase over the whole domain for the SOAmod scheme. These increases are most pronounced over Scandinavia for SOAvbs and for central Europe and Scandinavia for SOAmod. The maximum change

happens in the summer period, reaching a maximum of 700% for SOAmod for areas around the British Isles and around 500% in central Europe, while the differences for SOA2p simulations only show a maximum of 70% and 200% increase for annual and summer averages respectively, for the same area. This fact might suggest that the increase of BSOA concentrations due to climate change might be highly underestimated in future scenarios.

Despite the strong regional variations in the concentrations simulated by different schemes (figure 6, column 1), the geo-

graphic shape of the differences between historic and future scenarios (figure 6, second column) stay similar for all schemes, showing a maximum in the band between North and Baltic Sea. It is important to keep in mind that some of these differences occur in areas with low concentrations of BSOA, which can lead to large relative changes despite of only small absolute ones. This mostly occurs in the oceanic regions of the domain. When a land-sea mask is used, the maximum changes occur on the British Isles, Scandinavian area and the central Europe.

Figure 6 also shows the spatial distribution of temperature increase is correlated with that of BSOA increases (for all the schemes). There is an exception for the Mediterranean area, where absolute temperatures are high, but the concentration of BSOA is low, mainly because biogenic precursors of BSOA are not emitted in this area.

## 5 Sensitivity of different schemes to temperature changes

Figure 7 shows the logarithm of normalized concentrations of BSOA for EUR and MED sub-domains plotted against temperature, for the summer period, using daily average values for each scheme for the five considered summer periods. Dashed lines correspond to linear least-square fits for historic simulations and full lines for future scenarios. BVOC emissions have been added to the plot as well. Normalization of the data has been done by a division by the average of each set of simulations, then the natural logarithm of this ratio is calculated. It is important to bear in mind that as mentioned before, for future scenarios the years with highest temperature and highest BSOA aerosol concentrations are chosen. For the historic scenarios the years with lowest temperature and lowest BSOA aerosol concentrations are chosen, which explains the high difference between historic and future simulations (figure 7). Before entering into the discussion around sensitivity to temperature changes, it is important to keep in mind that the circulation patterns can change between the historic and future periods. Although the average of 5 years of simulations likely filters out part of the noise in these patterns, this could also affect BSOA concentrations in addition to temperature changes, especially over the Mediterranean remote with respect to sources.

As seen in figure 7, there is a high correlation between BVOC emissions and temperature throughout all the seasons (shown here for summer), showing an exponential behavior with temperature. The relationship between BVOCs and temperature is reported also for the Mediterranean basin, though the emissions of these species in this area are negligible. Accordingly, the correlation is lower over this area.

When looking at the different schemes, the regression lines show some differences for the future period. Interestingly SOAmod shows a slope rather similar to that of BVOC, while slopes are lower for the SOA2p and SOAvbs. Thus, for SOAmod, the temperature induced increase in BVOC fully affects BSOA. In contrast, for SOA2p and SOAvbs, less BSOA is formed with a temperature increase as could be expected from the correspondence of the temperature with BVOC emissions. This negative sensitivity of BSOA formation normalized by BVOC emissions is due to a shift of SVOC species to the gas phase for increasing temperature, as has been mentioned before. Apparently, this effect is much less pronounced or absent for SOAmod, probably because it includes, contrary to the other two schemes, formation of non-volatile SOA. Indeed, the SOAmod scheme shows 80% of the OA mass in the non-volatile bins, while the SOAvbs and the SOA2p schemes only shows respectively around 10% and 20% in these bins. These results suggest that the parameterization of OA schemes might lead to different sensitivity in prediction of the OA load with respect to the variations in the temperature. The same tendencies are observed for the historic period; however they show a lower intensity because of the lower general temperature ranges.

## 6 Conclusions

In this study, we presented the effect of different OA simulation schemes on future aerosol projections due to climate change. For this purpose, three schemes have been used, a molecular single-step oxidation scheme (SOA2p), a standard VBS scheme with anthropogenic SVOC aging only (SOAvbs) and a modified VBS scheme containing functionalization, fragmentation and formation of non-volatile SOA for all SVOC species (SOAmod). These schemes were evaluated for the European region for the year 2013. Although showing differences with observations, each one of OA schemes performs within accepted error ranges.

Since VBS schemes are numerically demanding, only 10 years of simulations could be performed for each scheme. In order to maximize the differences between future and historic simulations, the RCP8.5 scenario was used. For the future scenarios, years where the temperature and the BSOA concentration were both at their maximum were chosen, while, for the historic simulations, 5 years with the lowest temperature and BSOA concentrations were selected. Indeed, climate change induced modifications were shown to affect especially the BSOA fraction of organic aerosol. Since BSOA contributes to an important degree to the total concentration of OA, the focus of this article is the evolution of BSOA concentrations in different schemes in future climatic projections.

The results show that the change in concentration indicated by the SOAmod scheme is stronger especially for summertime, showing a difference of +113%, +155% and +262% for SOA2p, SOAvbs and SOAmod respectively, for the European area. These changes are mostly due to increased BSOA formation, which is the major SOA fraction during summer. Previous studies investigated the changes in BSOA concentrations for future scenarios using a two-product scheme for the simulation of SOA. Thus, our suggestion is that the relative variation in SOA concentrations predicted with such schemes might be underestimated.

The reason for the augmentation of BSOA concentrations due to climate change in future scenarios is because of the high dependency to BVOC emissions (which are major precursors of the formation of BSOA in summer/warm periods) to temperature. In a future climate, with the increase of temperatures values, the emissions of BVOCs might increase, and in our case, they were predicted to increase by 88% for terpenes and 82% for isoprene (over the European domain). The effect on BSOA formation is tempered by the fact that higher temperatures favor the transition of semi-volatile organic material in the gas phase. This effect is much more pronounced for SOA2p and the SOAvbs schemes than for the SOAmod scheme, which is the only scheme in our study including aging of biogenic SVOCs and the formation of non-volatile SOA. The sensitivity of the SOAmod scheme to temperature is the lowest, and its relation to BVOC emissions the most linear.

The differences were analyzed for the Mediterranean area as well, since organic aerosol and BSOA are transported to this area from continental Europe. While the concentrations in the Mediterranean and changes for future climate are lower for BSOA in general compared to the European area, the changes for this region are stronger in the SOAmod scheme as well (76%, 75% and 127% for SOA2p, SOAvbs and SOAmod respectively for summer).

In conclusion, our study suggests that the BSOA concentrations changes are highly sensitive to climate change and the scheme used for their simulation. The changes reported until now for future scenarios are highly uncertain, both on absolute and on relative scale. On a relative scale, these changes might be higher with OA schemes that include formation of non-volatile SOAs (up to a factor of two).

Future work is necessary in still developing more accurate organic aerosol schemes, not only in terms of absolute concentrations simulated, but also with respect to their temperature sensitivity. The three schemes used in this study can accurately simulate the concentrations of OA each for a specific season and for a specific region, while none of the schemes seem to be able to do so for the whole domain. Therefore, more research is necessary in order to develop OA simulation schemes that are able to represent the concentrations of OA accurately and the temperature sensitivity of this species on a regional scale.

*Data availability.* Access to the data used in this article is restricted to registered users of the ChArMEx project. The data are available on the project website (http://mistrals.sedoo.fr/ChArMEx/, last access: 17 June 2019) and it should be used following the data and publication policies of the ChArMEx project;http://mistrals.sedoo.fr/ChArMEx/Data-Policy/ChArMEx_DataPolicy.pdf. The data used for the scheme validation part of the study was downloaded from EBAS (http://ebas.nilu.no/) .

*Author contributions.* ArC, AuC and MB designed the experiment. ArC and AuC performed the simulations, and ArC carried out the post-processing of aforementioned simulations. Article reduction was performed by ArC, and all authors contributed to the text, interpretation of the results and review of the article.

*Competing interests.* The authors declare that they have no conflict of interest.

*Special issue statement.* This article is part of the special issue "CHemistry and AeRosols Mediterranean EXperiments (ChArMEx) (ACP/AMT
inter-journal SI)". It is not associated with a conference.

*Acknowledgements.* This research has received funding from the French National Research Agency (ANR) projects SAF-MED (grant ANR-15 12-BS06-0013). This work is part of the ChArMEx project supported by ADEME, CEA, CNRS-INSU and Météo-France through the multidisciplinary program MISTRALS (Mediterranean Integrated Studies aT Regional And Local Scales). The work presented here received support from the French Ministry in charge of ecology. This work was performed using HPC resources from GENCI-CCRT (Grant
2018-A0030107232). R. Vautard is acknowledged for providing the WRF/IPSL-CM5-MR Cordex simulations, and D. Hauglustaine and S. Szopa are acknowledged for providing the INCA simulations. Z. Klimont is acknowledged for providing ECLIPSE-v4 emission projections. The thesis work of Arineh Cholakian is supported by ADEME, INERIS (with the support of the French Ministry in charge of Ecology), and via the ANR SAF-MED project. Giancarlo Ciarelli was supported by ADEME and the Swiss National Science Foundation (grant no. `P2EZP2_175166`).

**Appendix A**

| Country | Station name | Longitude | Latitude | Altitude | Type | Duration | Resoltuion |
|---|---|---|---|---|---|---|---|
| Switzerland | CH0001G | 7,99 | 46,55 | 3578,0m | PM1 | 9mo | 1h |
| Switzerland | CH0002R | 6,94 | 46,81 | 489,0m | PM2.5 | 1yr | 1d |
| Switzerland | CH0005R | 8,46 | 47,07 | 1031,0m | PM2.5 | 1yr | 1d |
| Switzerland | CH0033R | 8,93 | 46,16 | 203,0m | PM1 | 18w | 1h |
| Cyprus | CY0002R | 33,06 | 35,04 | 520,0m | PM2.5 | 1yr | 1d |
| Czech republic | CZ0003R | 15,08 | 49,57 | 535,0m | PM2.5 | 1yr | 1d |
| Germany | DE0002R | 10,76 | 52,80 | 74,0m | PM2.5 | 1yr | 1d |
| Germany | DE0003R | 7,91 | 47,91 | 1205,0m | PM2.5 | 1yr | 1d |
| Germany | DE0007R | 13,03 | 53,17 | 62,0m | PM2.5 | 1yr | 1d |
| Germany | DE0008R | 10,77 | 50,65 | 937,0m | PM2.5 | 1yr | 1d |
| Germany | DE0044R | 12,93 | 51,53 | 86,0m | PM10 | 1yr | 1d |
| Spain | ES0001R | -4,35 | 39,55 | 917,0m | PM2.5 | 1yr | 1d |
| Spain | ES0009R | -3,14 | 41,28 | 1360,0m | PM2.5 | 1yr | 1d |
| Spain | – | 3,03 | 39,84 | 15,0m | PM1 | 3mo | 1h |
| Spain | ES1778R | 2,35 | 41,77 | 700,0m | PM1 | 6mo | 1h |
| Finland | FI0050R | 24,28 | 61,85 | 181,0m | PM1 | 9mo | 1h |
| France | FR0009R | 4,63 | 49,90 | 390,0m | PM2.5 | 1yr | 6d |
| France | FR0013R | 0,18 | 43,62 | 200,0m | PM2.5 | 1yr | 6d |
| France | FR0030R | 2,95 | 45,77 | 1465,0m | PM2.5 | 1yr | 2d |
| France | – | 9,38 | 42,97 | 520,0m | PM1 | 3mo | 1h |
| France | – | 2,15 | 48,71 | 156m | PM2.5 | 1yr | 1d |
| Greece | GR0002R | 25,67 | 35,32 | 250,0m | PM10 | 1yr | 1d |
| Ireland | IE0031R | -9,90 | 53,33 | 10,0m | PM1 | 41w | 1h |
| Italy | IT0004R | 8,63 | 45,80 | 209,0m | PM2.5 | 1yr | 1d |
| Netherlands | NL0644R | 4,92 | 51,97 | 1,0m | PM2.5 | 1yr | 1d |
| Norway | NO0002R | 8,25 | 58,39 | 219,0m | PM1 | 1yr | 1w |
| Norway | NO0039R | 8,88 | 62,78 | 210,0m | PM1 | 1yr | 1w |
| Norway | NO0056R | 11,08 | 60,37 | 300,0m | PM2.5 | 1yr | 1d |
| Poland | PL0005R | 22,07 | 54,15 | 157,0m | PM10 | 1yr | 1d |
| Sweden | SE0011R | 13,15 | 56,02 | 175,0m | PM10 | 1yr | 1d |
| Sweden | SE0012R | 17,38 | 58,80 | 20,0m | PM10 | 1yr | 1d |
| Slovenia | SI0008R | 14,87 | 45,57 | 520,0m | PM2.5 | 1yr | 1d |

**Table 2.** Names and other information about the stations used in scheme validation part of this study.

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
