# Peer review of "Biogenic SOA sensitivity to organic aerosol simulation schemes in climate projections"

_Atmospheric Chemistry and Physics, 2019_

## Referee Comment (RC1) · Anonymous Referee #2 · 9 Jul 2019

In this manuscript, Arineh Cholakian et al. apply the CHIMERE CTM to the European domain and the Mediterranean region. They present the differences in BSOA by comparing 5 years in the past (historical) and 5 years in the future. The choice of the years aimed to maximize the differences between future and historic simulations regarding the change in temperature.

The authors explored three schemes (i) a molecular single-step oxidation scheme, (ii) a standard VBS scheme with anthropogenic SVOC aging only and (iii) a modified VBS scheme containing functionalization, fragmentation and formation of non-volatile SOA for all SVOC species. The year 2013 was used in order to evaluate the schemes for the European region.

Major comments

For the entire manuscript it is not clear if the authors refer to $PM_{10}$ or $PM_{2.5}$ BSOA mass concentration. Please clarify.

P7 line 15. The observations are filter-based or online measurements; $PM_{10}$, $PM_{2.5}$, $PM_{1}$?

P7 line 24. The authors should explain the correlation. A coefficient of determination ($R^2$) should be used instead of correlation coefficient (R).

P7 line 25-27. "The three schemes perform reasonably well according to the criteria introduced by Boylan and Russell (2006), with the values for all the schemes falling into in zone 1 for both mean fractional bias and mean fractional error." Please provide more information about the criteria by Boylan and Russell (2006). Provide more information about Taylor diagram.

P7 line 27-29. "Each one of the schemes performs better for a specific period; modified VBS in summer, CHIMERE standard scheme during winter, and the standard VBS scheme showing average performance during the whole year." This result is not clear in Figure 3 or 4. Please provide which statistical metric is used for this statement.

Figures 4 and 5. The authors, currently present together European sub-domain and Mediterranean Sea sub-domain. It would be better if they split the two regions, as Mediterranean is discussed in the next chapter. In Fig.4 keep only a, b and c, and add from Fig. 5 the EUR-related figures. The same applies for MED-related ones, keep d, e and f and MED form Fig.5.

P8 lines14-15. Please provide which correlation the authors are referring to.

P9 line 3-4. "We address results for BSOA, as it makes the major contribution to OA during summer (between 40 and 78% for different schemes in the historic scenario)" In which figure is this shown?

P9 lines 5-6. "while for SOA2p an increase of +94% is calculated, this percentage raises to +135% for SOAvbs and +189% for SOAmod" These numbers do not correspond to Fig.4 c1 and summer season.

P10 lines 5-8. The numbers given in the manuscript are not consistent to figure 4 c.1. Please confirm the right one.

P10 lines 11-13. The numbers given in the manuscript are not consistent to figure 4 c.2. Please confirm the right one.

P10 line 25. How were calculated these percentages? How are they linked to Fig. 5?

P10 line 25-26. "SOA2p indicates a higher increase in nonfossil contribution compared to other schemes." This cannot be stated unless the actual concentrations are shown. Please clarify if the authors are referring to the percentage of the increase.

P11 line 4-5. From Fig. 5 HOA is not that much different between SOA2p and SOAmod schemes. In contrast, SVOOA is a lot higher in the SOA2p than SOAmod. Please clarify if the authors are referring to actual concentrations. If yes, please provide a figure.

P11 lines 5-6. A more in depth explanation of why the LVOOA is underestimated is needed.

A suggestion to the authors would be the addition of a distribution of simulated organic aerosol (OA) in volatility bins for each scheme.

P11 lines 8-9. "The contribution of HOA in future scenarios becomes less compared with historic simulations, probably since more BSOA formation happens in future scenarios". Please verify if this applies for the actual concentration of HOA, as from Fig.5 the HOA contribution slightly changed between the historic and future simulations.

P12 lines 5-7. "However, the predicted increase for the future is higher for SOAvbs and SOAmod (figure 6, second column), reaching a maximum of 300% increase for the SOAmod scheme." These numbers are not shown in Fig. 6 column 2. Is the second column referring to annual or summer period?

P12 Figure 6. Please clarify the second column if it is from summer period. Add the summer indication also in the caption for columns first and third. A suggestion would be to add the corresponding figure 6 for the annual simulation.

P13 lines 22-24. The authors state that BSOA in SOAmod scheme presents the highest change. This is not true for the actual concentrations of BSOA, as SOA2p shows the highest absolute change. Also, regarding the relative change this applies only during summer period.

P13 lines 29-30. The statement is not true according to Fig. 5. The fossil sources are the major contributors only in the case of SOAvbs.

P14 Figure 7. Please use the same axis for the two cases. Also, it is not clear where the cycles and cubes are in the figures. The use of the coefficient of determination ($R^2$) instead of correlation coefficient (R) is advised.

P15 line 22. Change the corresponding numbers according to the main manuscript.

P16 line 4. Change the corresponding numbers according to the main manuscript. Currently are not consistent.

A clarification between absolute and relative changes and annual and summer period should be made for the whole manuscript.

Minor comments:

P2 line 12. Remove commas after the dots in reference.

P2 line 31. Remove comma after the dot in reference.

P2 line 32. Define CHIMERE.

P2 line 34. Remove all the dots in $\mu g.m^{-3}$

P2 line 35. Remove comma after the dot in reference.

P5 line 12. Remove commas after the dots in reference.

P5 line 31. Add a dot after Cholakian.

P7 line 12. Define ECMWF.

P7 line 13. Define MEGAN.

P8 lines 11-12. Remove dots in $molecules.cm^{2}$

P8 Figure 3. Check the order of the months. June-July-August are before March-April-May. Please magnify the table. Replace bials with bias.

P9 Figure 4. Line2. Please add next to historic simulations, also future simulations.

P10 line 12. Change august to August.

P12 Figure 6. Remove dots from units.

P15 line 6. Change VBS mod to SOAmod.

---

## Referee Comment (RC2) · Anonymous Referee #1 · 12 Jul 2019

The paper investigates three aspects of biogenic SOA 1) impact of climate change 2) sensitivity to SOA formation scheme. 3) regional differences, in particular effects over Mediterranean sea Although the first question has been addressed before, the combination with assessment with different SOA formation schemes and the huge sensitivities to them that were found make the paper interesting. The applied methods are sound and the outcomes are interpreted well.

The fact that several aspects are addressed makes it however more difficult to present the results. This is already is also reflected in the title, which tries to summarize this but is difficult to understand and not completely correct. The order of presenting the

results could also be improved, e.g. present figure 6 earlier in the manuscript to give the reader an idea of the gradients and order of magnitude, and going into validations and more detailed analyses. Also, since absolute concentrations are rather different for the three schemes, comparing relative differences is on the one hand a respected method, but one should be careful especially when concentrations are low and small absolute differences are exaggerated. Structure of section 4 is not consistent with subchapters: would be easier for the reader to present section 4.3 after presentation of Europe and Mediterranean. Sometimes the authors should be more precise. Below, detailed comments on the various parts of the paper will be given. The motivation to focus on the summer and on biogenic SOA is there but could also be made a bit more prominent. Detailed comments

L1 Organic aerosol (OA). . .introduce abbreviation here.

L6 The differences between three different schemes to simulate OA are explored. These schemes are. . . 1) a molecular scheme, 2) a standard. . ..

L12 These changes a re largest over the summer period. . .

L17: Absolute concentrations are different: move this sentence to l 10, before addressing the relative change, also quantify (molecular scheme gives twice as much OA as VBS scheme)

P2 l 8 e.g. temperature change, land use changes and CO2 inhibition (Heald..)

P2 l 19 BVOC emissions have been quantified. . ..mainly to assess the future evolution of. . .. Statement that it is only emerging does not reflect the date of the papers cited, and they are of the same period as the Arneth paper.

P3l10 Impact of climate change is different per region since emissions and atmospheric composition are different per region, and depend on sensitivity of scheme to temperature changes. The description of the thermodynamics are different between the scheme. The current sentence is not precise and confusing

P4l6 More information on the modelling framework in the current study is provided in..

P5 l2: the domain covers latitudes 30-70°N and longitudes 40W-60E

P5 l 32: What about the performance of SOA2p in the Mediterranean, as compared to SOAvbs and SOAmod?

Figure 2:averaged pver 70 years of RCP scenarios (2031-2100) and averaged over 30 years for historic simulations. A-c mentioned in text, not indicated in figure

P6: l10. For the perspective it would be good to mention that winter OA concentrations in cold episiodes are higher that summer OA concentrations (Table 3)

P7 l25-26: would be better to stick to SOAvbs and SOAmod convention as defined on p 5. Differences in correlation are small.

Table in figure 3 is nearly unreadable, shoud be put as a text table outside the figure for readability. Also fonts of figure tick labels too small.

P8 l9 : change quite a bit: be more precise

P8 l 11 annual average "historical"..., annual average historical .... Unit is per cm2 I assume, this is not what it states. Correct woud be molecules cm-2 yr-1.

P8 l17-20: Leave out sentence : for the Mediterranean region, there are no local emissions..... The following sentence is more clear by itself.

Fig 4 Quality of graphics is poor, tick labels unreadable, too small

P9l3 Important sentence, could be combined with your reference to Cholakian (2019) in the section on choice of years, and maybe included in the introduction, depending on how you see this as a motivation and how this is the result of the present study. Would help the reader to get this statement very clear in an earlier part of the paper.

P10 l3 The statement can be related to seasonality in biogenic emissions.

P10 l16: What do yo mean by distribution of origins and volatility bin aspects?
P11 l8: relative contribution of HOA becomes less, since in your scenario you did not change anthropogenic emissions, but more BSOA is formed.

P12 Fig 6: State in caption that these are summer averages.

P12 l8 The maximum change is found in the summer period, reaching a maximum of ...These large differences are for areas with very low concentrations, so the small increase in absolute numbers is blown up. A few words should be devoted to this aspect, to indicate where the most relevant increases are found. Although Baltic sea and North Sea are identified as region with larges changes, I think that the relevance is larger for central Europe where a small relative increase implies substantially higher absolute SOA concentrations. When you put section 4.3 after section 4.4 you could easily include the Mediterranean as a focus area in this discussion and make it the ultimate summary.

P13 l 15-18: Part on PM10 not relevant for this paper

P13 l 27 : difference, not change

P14: Figure 7: differnces between dots and squares only visible when enlarging on computer screen, not when one prints the paper.

P14 section 5: In climate projections, not only the temperature changes but also the circulation patterns (and even differently for different global climate models). This might be the reason for the different directions in the lines for the two periods. Also, when looking at a smaller domain, this change in circulation may become more relevant instead of averaging it out over a large domain. In particular over the Mediterranean without sources of isoprene/terpenes, changes in transport patterns are important.

P15 l 4 correpsondence of what to what?

P16 Conclusionss: reasoning could be slightly more precise. Your results show that BSOA changes due to climate change are highly sensitive to the SOA scheme used, and that none of th BSOA schemes here matches the observations, which shows the

importance of further development of more accurate SOA schemes. When the SOA scheme is truly accurate, a good temperature dependence would be implicit, since you would like to use it over cold areas like Scandinavia as well as warmer areas like the Mediterranean.
* * *

---

## Referee Comment (RC3) · Anonymous Referee #3 · 13 Jul 2019

In this work, Cholakian et al. used three different organic aerosol simulation schemes in order to identify how they impact the calculated OA load on future climate projections. They found significant differences on the calculated biogenic SOA projections over Europe between the three OA schemes; highlighting the uncertainties that still exist on OA calculations. This study is of definite interest to the organic aerosol modeling community by contributing towards the understanding of the source of uncertainty between OA schemes (e.g., highlighting the role of temperature sensitivity). Overall, the manuscript is very well written and the presentation is clear. Therefore, I recommend this study for publication. Below are a few minor comments to be considered prior to publication.

[Figure]

Specific comments:

1. Title: I believe the manuscript focuses a lot on biogenic SOA, therefore is better to replace the general term "particulate matter" with biogenic SOA. Furthermore, the manuscript presents the sensitivity of BSOA concentrations on the OA scheme used and not vice versa as the title implies. I suggest to consider revising the title.

2. Page 1 lines 7-8: the word "formation" is unnecessarily repeated two times in the sentence

3. Page 1 lines 7-8: This sentence is not clear. I assume you men the temperature differences.

4. Page 2 line 1: Tsimpidi et al. (2017, doi: 10.5194/acp-17-7345-2017) is also a nice recent study that emphasizes the large uncertainty of OA formation.

5. Page 2 line 3: Lelieveld et al. (2015, doi:10.1038/nature15371) also highlight the adverse effects of OA on human health due to their increased toxicity.

6. Page 2 lines 22-25: The scheme of Pankow (1994, doi: 10.1016/1352-2310(94)90093-0) should be included in the discussion here

7. Page 4 1st pargraph: More information is needed for the simulations conducted by the global models and WRF (e.g., which RCPs were used, which is the suimulation period, etc. ?) Furthermore, are all the links between the models offline?

8. Page 5 lines 18-21: It has been also shown that the aging of BSOA does not lead to any net changes on its mass concentration due to a balancing effect between fragmentation and functionalization (Murphy et al., 2012, doi: 10.5194/acp-12-10797-2012)

9. Page 5 lines 22: Actually, the standard VBS scheme assumes that fragmentation and functionalization processes result in a net average decrease in volatility for SOA. Therefore, even if it does not simulate explicitly the fragmentation process, it has taken

into account its effects on the SOA volatility changes.

10. Page 5 lines 27: Can you briefly discuss the main differences between the standard and the modified VBS schemes (e.g., aging rate constants, changes on volatility and oxygen atoms added after each reaction step, etc.?)

11. Page 7 lines 22: Can you comment on why all schemes significantly fail to reproduce the observed OA concentrations during winter?

12. Page 8 lines 11: Add "the" before "average". Furthermore, the emission units do not seem correct. You need "amount time-1 area-1". It would be better to report the emissions in Tg/yr for the whole domain.

13. Page 9, Figure 4: The figure caption states that these are BSOA but the figure legend has SOA (i.e., SOA2p, SOAvbs, SOAmod)

14. Page 9 lines 6-9: I found this sentence long and confusing

15.Page 10 lines 12: use capital A for August

———————————————————

---

## Author Comment (AC1) · 30 Sep 2019

The authors thank the three referees for their thorough and pertinent reviews, which certainly allow us to improve the paper. Below we provide answers to points raised by each of the referees.

In this response, bold black parts are direct extracts of referee comments, blue italic parts are changes made in the article and black normal texts are answers/explanations on each comment made by the referees.

**Referee 1 comments:**

We would like thank this referee for their pertinent remarks, in the section that follows we try to answer every question/comment raised by this referee.

**Referee 1 general comments:**

1. **The fact that several aspects are addressed makes it however more difficult to present the results. This is already is also reflected in the title, which tries to summarize this but is difficult to understand and not completely correct.**

This comment has been addressed both in the title and in the text. The title has been modified to:

*Biogenic SOA sensitivity to organic aerosol simulation schemes in climate projections*

2. **The order of presenting the results could also be improved, e.g. present figure 6 earlier in the manuscript to give the reader an idea of the gradients and order of magnitude, and going into validations and more detailed analyses.**

This comment has been taken into account partly, modifying the order of presenting the results in section 4, however figure 6 has been kept where it was. Section 4 has been modified according to the comment number 4 of this referee. We chose to keep figure 6 where it is since the discussion for figures 4 and 5 come before figure 6 and wed like to have a more general (averaged on sub-domains) discussions before entering into the regional changes in 2D images.

3. **Also, since absolute concentrations are rather different for the three schemes, comparing relative differences is on the one hand a respected method, but one should be careful especially when concentrations are low and small absolute differences are exaggerated.**

Yes, this comment has been taken into account and will be specified in answering the detailed remarks below.

4. **Structure of section 4 is not consistent with subchapters: would be easier for the reader to present section 4.3 after presentation of Europe and Mediterranean.**

Yes, we have modified the order of subsections in section 4 in order to put the spatial analysis after the discussion of the two sub-domains.

5. **Sometimes the authors should be more precise. Below, detailed comments on the various parts of the paper will be given.**

The detailed comments have been taken into account thoroughly and explanations have been given for each one below.

6. **The motivation to focus on the summer and on biogenic SOA is there but could also be made a bit more prominent.**

Some phrases have been added (as per suggestions by the referee #1 and referee #3) in order to address this comment. They have been mentioned in the detailed comments below.

**Referee 1 Detailed comments:**

1. **L1 Organic aerosol (OA). . .introduce abbreviation here.**

OA has been added to this sentence.

2. **L6 The differences between three different schemes to simulate OA are explored. These schemes are. . . 1) a molecular scheme, 2) a standard. . ..**

Modified.

3. **L12 These changes are largest over the summer period. . .**

Modified.

4. **L17: Absolute concentrations are different: move this sentence to l 10, before addressing the relative change, also quantify (molecular scheme gives twice as much OA as VBS scheme)**

The sentence was moved higher, before addressing the relative changes, the following part was added as well:

> *Absolute concentrations between different schemes are also different, the molecular scheme showing the highest concentrations between the three schemes.*

5. **P2 l 8 e.g. temperature change, land use changes and CO2 inhibition (Heald..)**

Modified.

6. **P2 l 19 BVOC emissions have been quantified. . ..mainly to assess the future evolution of. . .. Statement that it is only emerging does not reflect the date of the papers cited, and they are of the same period as the Arneth paper.**

That is true. The phrase has been modified to the following:

> *It is mainly to assess the future evolution of tropospheric ozone that BVOC emissions have been quantified at global scale in chemistry-climate projections (Arneth et al., 2010). Their importance for organic aerosol chemistry has also been considered in global and regional scale atmospheric models (Maria et al., 2004; Tsigaridis et al., 2007; Heald et al., 2008b), but to a lesser degree.*

7. **P3l10 Impact of climate change is different per region since emissions and atmospheric composition are different per region, and depend on sensitivity of scheme to temperature changes. The description of the thermodynamics are different between the scheme. The current sentence is not precise and confusing**

The phrase has been modified to the following:

> *Differences induced by different schemes are also expected to vary regionally, depending on the concentration ranges encountered and ranges and changes in meteorological parameters.*

8. **P4l6 More information on the modelling framework in the current study is provided in..**

Modified.

9. **P5 l2: the domain covers latitudes 30-70◦N and longitudes 40W-60E**

Modified.

10. **P5 l 32: What about the performance of SOA2p in the Mediterranean, as compared to SOAvbs and SOAmod?**

A detailed comparison of these schemes in two Mediterranean sites (in Corsica and Mallorca) was performed and published in Cholakian et al, 2018. The results show that the SOA2p shows a high overestimation in the simulated concentrations compared to the two VBS based schemes, while the other two both perform quite well for the simulation of the concentration. However, when it comes to the simulation of oxidation state and origins of the observed OA, the modified VBS scheme performs better than the standard VBS scheme (which overestimates the fossil contribution and doesn't age particles as much as it should).

11. **Figure 2:averaged pver 70 years of RCP scenarios (2031-2100) and averaged over 30 years for historic simulations. A-c mentioned in text, not indicated in figure**

The information has been added to the figure legend.

12. **P6: l10. For the perspective it would be good to mention that winter OA concentrations in cold episiodes are higher that summer OA concentrations (Table 3)**

It is in the data, but the passage here is about BSOA being the major source of OA in the summer months, since the study revolves around BSOA concentrations.

13. **P7 l25-26: would be better to stick to SOAvbs and SOAmod convention as defined on p 5. Differences in correlation are small.**

The phrase has been modified to:

> *The correlation between observed and simulated OA concentrations for different schemes are the highest for the SOAmod, and lowest for the SOAvbs in most seasons; it should also be noted that the difference between the correlations seen for each scheme are rather small (difference of below 0.05).*

14. **Table in figure 3 is nearly unreadable, shoud be put as a text table outside the figure for readability. Also fonts of figure tick labels too small.**

The tables on the side of the image have been moved to a separate table in the text. The size of the ticks has been modified and some general description has been added in the caption to describe the taylor diagram.

15. **P8 l9 : change quite a bit: be more precise**

The intensities of the changes have been quantified in the lines that come below this phrase. This sentence is meant as an introductory phrase to enter into the explanation about the intensity of these changes.

16. **P8 l 11 annual average "historical". . ., annual average historical . . .. Unit is per cm2 I assume, this is not what it states. Correct woud be molecules cm-2 yr-1.**

The historical has been changed to historic and the units have been modified.

**17. P8 l17-20: Leave out sentence : for the Mediterranean region, there are no local emissions. . ... The following sentence is more clear by itself.**

The paragraph was changed to:

*For the Mediterranean region, there are no local biogenic emissions included in the model.*

**18. Fig 4 Quality of graphics is poor, tick labels unreadable, too small**

The size of the figure has been increased; the labels have been changed as well.

**19. P9l3 Important sentence, could be combined with your reference to Cholakian (2019) in the section on choice of years, and maybe included in the introduction, depending on how you see this as a motivation and how this is the result of the present study. Would help the reader to get this statement very clear in an earlier part of the paper.**

The sentence was moved to the choice of years section. The following sentences were added to the abstract and the conclusion:

*The study focuses on BSOA since the contribution of this fraction of BSOA is more important in both historic and future scenarios (40 to 78\% for different schemes in historic simulations).*

*Since BSOA contributes to an important degree to the total concentration of OA, the focus of this article is the evolution of BSOA concentrations in different schemes in future climatic projections.*

**20. P10 l3 The statement can be related to seasonality in biogenic emissions.**

That is correct, the following phrase was added to the aforementioned line to highlight that fact:

*Highest relative changes occur for august for all schemes (+133\%, +168\% and +333\% for SOA2p, SOAvbs and SOAmod respectively) resulting from the seasonal profile of BVOC emissions.*

**21. P10 l16: What do you mean by distribution of origins and volatility bin aspects?**

The sentence sounds a bit ambiguous, what was meant is each scheme has its own way of simulating the origins of formed OA (because of difference in aerosol formation reactions) as well as the volatility of simulated OA. For example, the VBS scheme has a bin-per-bin distribution. The sentence is modified to:

*Since the schemes behave differently both in contribution of different origins in the formation of OA as well as volatility distribution of OA, it is interesting to compare these two aspects in the tested schemes.*

**22. P11 l8: relative contribution of HOA becomes less, since in your scenario you did not change anthropogenic emissions, but more BSOA is formed.**

That is what was meant by the second part of the phrase 'probably since more BSOA formation happens in future scenarios'. More explanation has been added to the phrase:

*The relative contribution of HOA decreases in all schemes since the anthropogenic emissions are kept constant in future simulations and the relative contribution of SOA increases in the future.*

**23. P12 Fig 6: State in caption that these are summer averages.**

The caption has been modified.

**24. P12 l8 The maximum change is found in the summer period, reaching a maximum of . . .These large differences are for areas with very low concentrations, so the small increase in absolute numbers is blown up. A few words should be devoted to this aspect, to indicate where the most relevant increases are found. Although Baltic sea and North Sea are identified as region with larges changes, I think that the relevance is larger for central Europe where a small relative increase implies substantially higher absolute SOA concentrations. When you put section 4.3 after section 4.4 you could easily include the Mediterranean as a focus area in this discussion and make it the ultimate summary.**

Putting section 4.4 before section 4.3 is quite logical, the modification has been made in the paper. In regards to smaller differences being amplified when looking at relative changes, this occurs mostly over the ocean, so a possible solution would have been using a land-sea mask. However, since the discussion is around the Mediterranean area, this idea was disregarded. Therefore, some sentences were added in order to address this fact:

*It is important to keep in mind that some of these differences occur in areas with low concentrations of BSOA, which can lead to large relative changes despite of only small absolute ones. This mostly occurs in the oceanic regions of the domain. When a land-sea mask is used, the maximum changes occur on the British Isles, Scandinavian area and the central Europe.*

**25. P13 l 15-18: Part on PM10 not relevant for this paper**

That is true, the phrase is meant to highlight the differences between the two different areas. The remark about $PM_{10}$ not being relevant to the paper is true, therefore the phrase was changed to the following:

*There are major differences between the concentrations of different aerosol components over the Mediterranean area compared to continental Europe.*

**26. P13 l 27 : difference, not change**

Modified.

**27. P14: Figure 7: differnces between dots and squares only visible when enlarging on computer screen, not when one prints the paper.**

This point was raised by referee #3 as well. The shapes of the points were changed in order to illustrate the differences more vividly.

**28. P14 section 5: In climate projections, not only the temperature changes but also the circulation patterns (and even differently for different global climate models). This might be the reason for the different directions in the lines for the two periods. Also, when looking at a smaller domain, this change in circulation may become more relevant instead of averaging it out over a large domain. In particular over the Mediterranean without sources of isoprene/terpenes, changes in transport patterns are important.**

This is true and it's a pertinent remark. Circulation patterns change between the two periods for each simulated year. However, since the average of 5 years of simulations are compared for each period, it is likely that the changes caused by the circulation pattern changes of a specific year are averaged out

especially since the years have been chosen in a way to represent low/high temperature maxima for historic/future periods respectively. The following phrase has been added in the article in order to mention this point:

> *Before entering into the discussion around sensitivity to temperature changes, it is important to keep in mind that the circulation patterns can change between the historic and future periods. Although averaging of 5 years of simulations likely filters out part of the noise in these patterns, this could also affect BSOA concentrations in addition to temperature changes, especially since the Mediterranean area since its remote with respect to sources.*

**29. P15 l 4 correpsondence of what to what?**

The sentence has been changed to:

> *In contrast, for SOA2p and SOAvbs, less BSOA is formed with a temperature increase as could be expected from the correspondence of the temperature with BVOC emissions.*

**30. P16 Conclusionss: reasoning could be slightly more precise. Your results show that BSOA changes due to climate change are highly sensitive to the SOA scheme used, and that none of th BSOA schemes here matches the observations, which shows the importance of further development of more accurate SOA schemes. When the SOA scheme is truly accurate, a good temperature dependence would be implicit, since you would like to use it over cold areas like Scandinavia as well as warmer areas like the Mediterranean.**

These are both good points, the following phrases were added to the conclusion to emphasize these remarks:

> *In conclusion, our study suggests that the BSOA concentrations changes are highly sensitive to climate change and the scheme used for their simulation. The changes reported until now for future scenarios are highly uncertain, both on absolute and on relative scale. On a relative scale, these changes might be higher with OA schemes that include formation of non-volatile SOAs (up to a factor of two).*

> *Future work is necessary in still developing more accurate organic aerosol schemes, not only in terms of absolute concentrations simulated, but also with respect to their temperature sensitivity. The three schemes used in this study can accurately simulate the concentrations of OA each for a specific season and for a specific region, while none of the schemes seem to be able to do so for the whole domain. Therefore, more research is necessary in order to develop OA simulation schemes that are able to represent the concentrations of OA accurately and the temperature sensitivity of this species on a regional scale.*

> **In this manuscript, Arineh Cholakian et al. apply the CHIMERE CTM to the European domain and the Mediterranean region. They present the differences in BSOA by comparing 5 years in the past (historical) and 5 years in the future. The choice of the years aimed to maximize the differences between future and historic simulations regarding the change in temperature. The authors explored three schemes (i) a molecular single -step oxidation scheme, (ii) a standard VBS scheme with anthropogenic SVOC aging only and (iii) a modified VBS scheme containing functionalization, fragmentation and formation of non -volatile SOA for all SVOC species. The year 2013 was used in order to evaluate the schemes for the European region.**

We would like to thank referee #2 for their pertinent and thorough comments, in the section that follows we will address every point raised by this referee.

**Referee 2 major comments:**

1. **A clarification between absolute and relative changes and annual and summer period should be made for the whole manuscript.**

The text is modified in order to reflect the nature of the comparison wherever necessary.

2. **For the entire manuscript it is not clear if the authors refer to PM10 or PM2.5 BSOA mass concentration. Please clarify.**

Since most of BSOA concentrations is included in the $PM_1$ fraction, this is the fraction that has been used in this article. In the scheme validation part of the paper the fraction used for the simulated BSOA corresponds to what the measurement is for (either $PM_1$ or $PM_{2.5}$, as explained in the next point). The following sentence has been added to the beginning of the "Analysis of the simulations":

> *It is important to keep in mind that from this section on, whenever BSOA concentrations are discussed the $PM_1$ fraction of this species has been used.*

Also, a table showing the country, longitude/latitude, duration, type and altitude of measurements used in this study has been added here. Station names refer to EBAS names, the measurement is not present in EBAS if a station name isn't given.

| Country | Station name | Longitude | Latitude | Altitude | Type | Duration | Resoltuion |
|---|---|---|---|---|---|---|---|
| **Switzerland** | CH0001G | 7,99 | 46,55 | 3578,0m | PM1 | 9mo | 1h |
| **Switzerland** | CH0002R | 6,94 | 46,81 | 489,0m | PM2.5 | 1yr | 1d |
| **Switzerland** | CH0005R | 8,46 | 47,07 | 1031,0m | PM2.5 | 1yr | 1d |
| **Switzerland** | CH0033R | 8,93 | 46,16 | 203,0m | PM1 | 18w | 1h |
| **Cyprus** | CY0002R | 33,06 | 35,04 | 520,0m | PM2.5 | 1yr | 1d |
| **Czech republic** | CZ0003R | 15,08 | 49,57 | 535,0m | PM2.5 | 1yr | 1d |
| **Germany** | DE0002R | 10,76 | 52,80 | 74,0m | PM2.5 | 1yr | 1d |
| **Germany** | DE0003R | 7,91 | 47,91 | 1205,0m | PM2.5 | 1yr | 1d |
| **Germany** | DE0007R | 13,03 | 53,17 | 62,0m | PM2.5 | 1yr | 1d |
| **Germany** | DE0008R | 10,77 | 50,65 | 937,0m | PM2.5 | 1yr | 1d |
| **Germany** | DE0044R | 12,93 | 51,53 | 86,0m | PM10 | 1yr | 1d |
| **Spain** | ES0001R | -4,35 | 39,55 | 917,0m | PM2.5 | 1yr | 1d |

| Spain | ES0009R | -3,14 | 41,28 | 1360,0m | PM2.5 | 1yr | 1d |
|---|---|---|---|---|---|---|---|
| Spain | -- | 3,03 | 39,84 | 15,0m | PM1 | 3mo | 1h |
| Spain | ES1778R | 2,35 | 41,77 | 700,0m | PM1 | 6mo | 1h |
| Finland | FI0050R | 24,28 | 61,85 | 181,0m | PM1 | 9mo | 1h |
| France | FR0009R | 4,63 | 49,90 | 390,0m | PM2.5 | 1yr | 6d |
| France | FR0013R | 0,18 | 43,62 | 200,0m | PM2.5 | 1yr | 6d |
| France | FR0030R | 2,95 | 45,77 | 1465,0m | PM2.5 | 1yr | 2d |
| France | -- | 9,38 | 42,97 | 520,0m | PM1 | 3mo | 1h |
| France | -- | 2,15 | 48,71 | 156m | PM2.5 | 1yr | 1d |
| Greece | GR0002R | 25,67 | 35,32 | 250,0m | PM10 | 1yr | 1d |
| Ireland | IE0031R | -9,90 | 53,33 | 10,0m | PM1 | 41w | 1h |
| Italy | IT0004R | 8,63 | 45,80 | 209,0m | PM2.5 | 1yr | 1d |
| Netherlands | NL0644R | 4,92 | 51,97 | 1,0m | PM2.5 | 1yr | 1d |
| Norway | NO0002R | 8,25 | 58,39 | 219,0m | PM1 | 1yr | 1w |
| Norway | NO0039R | 8,88 | 62,78 | 210,0m | PM1 | 1yr | 1w |
| Norway | NO0056R | 11,08 | 60,37 | 300,0m | PM2.5 | 1yr | 1d |
| Poland | PL0005R | 22,07 | 54,15 | 157,0m | PM10 | 1yr | 1d |
| Sweden | SE0011R | 13,15 | 56,02 | 175,0m | PM10 | 1yr | 1d |
| Sweden | SE0012R | 17,38 | 58,80 | 20,0m | PM10 | 1yr | 1d |
| Slovenia | SI0008R | 14,87 | 45,57 | 520,0m | PM2.5 | 1yr | 1d |

**3. P7 line 15. The observations are filter-based or online measurements; PM10, PM2.5, PM1?**

The observations are mostly $PM_{2.5}$, in some cases $PM_1$. None of the sites include $PM_{10}$ measurements. For each case ($PM_{2.5}$ or $PM_1$) the corresponding fraction from the simulations have been used. The following phrase has been added to the paper to include this explanation:

> *The used measurements are mostly $PM_{2.5}$, in some cases $PM_1$. None of the sites include $PM_{10}$ measurements. For each type of measurement ($PM_{2.5}$ or $PM_1$) the corresponding fraction from the simulations have been used.*

**4. P7 line 24. The authors should explain the correlation. A coefficient of determination (R2) should be used instead of correlation coefficient (R).**

The R values have been all modified to $R^2$.

**5. P7 line 25-27. "The three schemes perform reasonably well according to the criteria introduced by Boylan and Russell (2006), with the values for all the schemes falling into in zone 1 for both mean fractional bias and mean fractional error." Please provide more information about the criteria by Boylan and Russell (2006). Provide more information about Taylor diagram.**

The aforementioned reference fixes a model performance goal for PM with a mean fractional bias (MFB) and a mean fractional error (MFE) smaller than equal to +50% and ±30%, respectively. The model performance criterium is achieved when the 2 aforementioned statistic metrics are less than or equal to +75% and ±60% respectively. The author explains that the performance goals presented in this reference are close to the best the model can achieve, while performance criteria are defined as an acceptable performance for the model. In other words, in order to show the accuracy of a model, the

performance criteria have to be met, while the performance goals show the optimal performance of the model. The following information and values of MFB and MFE have been added to the paper:

*The goal for these two metrics according to the aforementioned reference is less than or equal to ±30% and +50% and the criteria is less than or equal to ±60% and +75% respectively. The MFB values for the three schemes are -19.7%, 16.5% and 26.9%, while MFE shows 47.9%, 51.1% and 47.2% for SOA2p, SOAvbs and SOAvbs respectively. Performance goals are met for nearly for all schemes, with a slight exceedance for MFE and the SOAvbs scheme, still meeting the criterium.*

The Taylor diagram summarizes several statistic information in one plot: The Correlation coefficient, root-mean-square (RMS) difference between observations and simulations and the standard deviation ratio can be seen. More information about the construct of this diagram is given in Taylor, 2001. The standard deviation is read on the radii of the quadrant, the correlation coefficient on the outside rim of the circle and the RMS on the demi-circles centered around the normalized standard deviation. For the Taylor diagram the following information has been added:

*The Taylor diagram in figure 3 shows the comparisons of different stations to simulations for each scheme (black for SOA2p, green for SOAmod and red for SOAvbs). The Taylor diagram summarizes several statistic information in one plot: The Correlation coefficient, root-mean-square (RMS) difference between observations and simulations and the standard deviation ratio can be seen. More information about the construct of this diagram is given in Taylor, 2001. As seen in this diagram, there is a high variability in the simulation of different stations, where some stations are better represented by the model than the others, which might be because of the geographical placement (altitude, types of emissions in said location, etc.) of stations or because of the nature of the station (urban, rural, etc.).*

6. **P7 line 27-29. "Each one of the schemes performs better for a specific period; modified VBS in summer, CHIMERE standard scheme during winter, and the standard VBS scheme showing average performance during the whole year." This result is not clear in Figure 3 or 4. Please provide which statistical metric is used for this statement.**

This fact is shown in the tables initially connected to figure 3 and now presented separately as table 1. Looking at table 1, it is seen that for example for the summer period, the SOAmod scheme shows the highest correlation of determination, while SOA2p shows the lowest bias for this season. For spring SOA2p shows the highest $R^2$, while SOAvbs shows the lowest bias. For winter and autumn, the performance of the schemes is quite similar. Annually, SOAvbs presents a similar correlation to SOA2p while showing the lowest bias in general. This has been added to the text as well:

*Looking at table 1, it is seen that for example for the summer period, the SOAmod scheme shows the highest correlation of determination, while SOA2p shows the lowest bias for this season. For spring SOA2p shows the highest $R^2$, while SOAvbs shows the lowest bias. For winter and autumn, the performance of the schemes is quite similar. Annually, SOAvbs presents a similar correlation to SOA2p while showing the lowest bias in general.*

7. **Figures 4 and 5. The authors, currently present together European sub-domain and Mediterranean Sea sub-domain. It would be better if they split the two regions, as Mediterranean is discussed in the next chapter. In Fig.4 keep only a, b and c, and add from Fig. 5 the EUR-related figures. The same applies for MED-related ones, keep d, e and f and MED form Fig.5.**

The two images have been modified according to this comment.

**8. P8 lines14-15. Please provide which correlation the authors are referring to.**

Here the correlation coefficient has been used. We have changed the correlation coefficients to correlation of determination in order to be consistent:

*The correlation of determination between historic isoprene and terpene emissions is 0.6 and 0.63 while this correlation is 0.65 and 0.57 for the future simulations.*

**9. P9 line 3-4. "We address results for BSOA, as it makes the major contribution to OA during summer (between 40 and 78% for different schemes in the historic scenario)" In which figure is this shown?**

This has not been shown in any of the figures, it is a value calculated using the simulations. However, since the majority of non-fossil OA comes from BSOA, this percentage can be implicitly seen in figure 5. The phrase "according to our simulation results for the historic period with differences schemes, not shown in figures" has been added to this statement in order to remove the confusion.

**10. P9 lines 5-6. "while for SOA2p an increase of +94% is calculated, this percentage raises to +135% for SOAvbs and +189% for SOAmod" These numbers do not correspond to Fig.4 c1 and summer season.**

These values correspond to annual averages, not the summer period. They are not shown in figure 4. It has not been mentioned in the manuscript to which period these values refer to, so we added this information:

*while for SOA2p an annually averaged increase of +94% is calculated, this percentage raises to +135% for SOAvbs and +189% for SOAmod for the same period.*

**11. P10 lines 5-8. The numbers given in the manuscript are not consistent to figure 4 c.1. Please confirm the right one.**

Scripting problem, the values in the figure 4 are correct, the values in the text have been changed to the right values:

*Summer shows the maximum relative increase (+113%, +155% and +262% for SOA2p, SOAvbs and SOAmod respectively) and winter the lowest one in all schemes (+31.1%, +26.2% and +20.5% for SOA2p, SOAvbs and SOAmod respectively). For autumn and spring SOA2p and SOAmod show similar and intermediate changes while SOAvbs shows higher differences (+59.6%/+40.3%, +79.9%/+60.0% and +57.3%/+50.0% for SOA2p, SOAmod and SOAvbs respectively for autumn/spring).*

**12. P10 lines 11-13. The numbers given in the manuscript are not consistent to figure 4 c.2. Please confirm the right one.**

Same issue as for the point above, the values have been changed:

*In July, when the BSOA concentration reaches its maximum, the percentage of change in the future is high as well (+115%, +151% and +243% for SOA2p, SOAvbs and SOAmod respectively). Highest relative changes occur for August for all schemes (+111%, +165% and +356% for SOA2p, SOAvbs and SOAmod respectively)*

**13. P10 line 25. How were calculated these percentages? How are they linked to Fig. 5?**

They represent the relative increase of the non-fossil part of the OA concentration between the future and historic simulations for each scheme on an annually averaged basis. The phrase has been modified to include the fact that the relative changes for annual averages are discussed:

> *All schemes show a relative increase in the contribution of non-fossil sources in the future on an annually averaged basis (10\%, 17\% and 22\% of increase for the non-fossil partition between future and historic simulations for SOA2p, SOAvbs and SOAmod respectively).*

**14. P10 line 25-26. "SOA2p indicates a higher increase in nonfossil contribution compared to other schemes." This cannot be stated unless the actual concentrations are shown. Please clarify if the authors are referring to the percentage of the increase.**

It refers to absolute values, but as the referee indicates the absolute values haven't been shown so, to avoid confusion, the phrase has been removed.

**15. P11 line 4-5. From Fig. 5 HOA is not that much different between SOA2p and SOAmod schemes. In contrast, SVOOA is a lot higher in the SOA2p than SOAmod. Please clarify if the authors are referring to actual concentrations. If yes, please provide a figure.**

The specified line and page do not correspond to the given sentence. Figure 5 shows relative changes for historic and future simulations. The text has been edited in order to assure that the fact that relative comparisons are meant to be discussed comes through more clearly.

**16. P11 lines 5-6. A more in-depth explanation of why the LVOOA is underestimated is needed. A suggestion to the authors would be the addition of a distribution of simulated organic aerosol (OA) in volatility bins for each scheme.**

While this is a great idea, the volatility bin distributions have been discussed in our previous work (Cholakian et al, 2018), and we would like to refrain from adding additional images only to show the formation of LVOOA to this article, especially since the oxidation state does not vary much for the Mediterranean area and the variations in oxidation state in the European area are already visible in figure 4-d. The requested image has been added to this response in order to include what the referee wanted to see. Also, for the sake of being more coherent, the images 4 and 5 have been modified to show HOA and SOA (sum of LVOOA and SVOOA) since SOAvbs does not form LVOOA particles and whatever is formed stays in the SVOOA section. he entire section has been modified, here are the modifications that pertain to this comment:

> *Figure 4-d shows a simplified distribution for the OA in different schemes: SOA and HOA (hydrocarbon-like organic aerosol) presenting the freshly emitted primary OA. Figure 4-d shows that the predicted distribution between HOA and SOA is different for the three schemes. SOA2p indicates a smaller contribution of SOA and a larger one from HOA compared to SOAvbs and SOAmod schemes. This is because POA emissions in SOA2p are considered non-volatile, while they are volatile in VBS schemes. The relative contribution of HOA decreases in all schemes in the future scenario, since the anthropogenic emissions are kept constant, and the concentration SOA increases. However, the decrease in the relative contribution of HOA is stronger for the SOAmod scheme, since it shows a higher relative increase in the formation of BSOA in future scenarios.*

We have also added the following phrase in section 5:

*Indeed, the SOAmod scheme shows 80\% of the OA mass in the non-volatile bins, while the SOAvbs and the SOA2p schemes only shows respectively around 10\% and 20\% in these bins.*

**17. P11 lines 8-9. "The contribution of HOA in future scenarios becomes less compared with historic simulations, probably since more BSOA formation happens in future scenarios". Please verify if this applies for the actual concentration of HOA, as from Fig.5 the HOA contribution slightly changed between the historic and future simulations.**

As it was mentioned by referee #1, the relative contribution of HOA becomes less since the relative contribution of LVOOA increases and the anthropogenic emissions don't change between historic and future simulations, by design of our study the absolute concentration of HOA does not change significantly, although some climatic response is seen for HOA as well. The following phrase has been added to the sentence:

*The relative contribution of HOA decreases in all schemes since the anthropogenic emissions are kept constant in future simulations and the relative contribution of SOA increases in the future.*

**18. P12 lines 5-7. "However, the predicted increase for the future is higher for SOAvbs and SOAmod (figure 6, second column), reaching a maximum of 300% increase for the SOAmod scheme." These numbers are not shown in Fig. 6 column 2. Is the second column referring to annual or summer period?**

All features shown in figure 6 are for the summer period. We have added a statement to make this clear in the figure caption, and we have also emphasized this fact in the text where we present figure 6. The phrase is changed to the following:

*reaching an average of 290% increase over the whole domain for the SOAmod scheme.*

**19. P12 Figure 6. Please clarify the second column if it is from summer period. Add the summer indication also in the caption for columns first and third. A suggestion would be to add the corresponding figure 6 for the annual simulation.**

Again, figure 6 is solely for the summer period. We have added that the values are for the summer period, and we have also emphasized this fact in the text where we present figure 6. An annual figure for the same image, although an excellent idea, adds another image to an article already overcharged by images, so the authors will like to respectfully keep the image that is shown and not add another one.

**20. P13 lines 22-24. The authors state that BSOA in SOAmod scheme presents the highest change. This is not true for the actual concentrations of BSOA, as SOA2p shows the highest absolute change. Also, regarding the relative change this applies only during summer period.**

That's true for the Mediterranean area, this has been changed in the text to the following:

*For BSOA relative changes, SOAmod still shows the largest relative change in the summer period compared to historic simulations (76\%, 75\% and 127\% for SOA2p, SOAvbs and SOAmod respectively), but the differences between schemes are less pronounced in the Mediterranean area.*

**21. P13 lines 29-30. The statement is not true according to Fig. 5. The fossil sources are the major contributors only in the case of SOAvbs.**

We agree. The phrase was modified to the following, to make a point about on larger fossile contributions in the med. area as compared to the European arean :

*As for the European are, the contribution of non-fossil sources in the future scenarios also increases in the Mediterranean area, but still contributes less than over the European area.*

**22. P14 Figure 7. Please use the same axis for the two cases. Also, it is not clear where the cycles and cubes are in the figures. The use of the coefficient of determination (R2) instead of correlation coefficient (R) is advised.**

The remark about using $R^2$ instead of R was taken into account and the figure was modified accordingly. The shapes of the points were changed to add a bit of a clearer view to the figure. The limits of the axis were not changed, since the two subdomains present very different axis limits both in terms of temperature and BSOA concentration/BVOC emissions which results in big empty spaces in both plots especially since the plots are on a logarithmic scale. The plot already being a bit complicated to read, we would like to keep the axis limits the way they are.

**23. P15 line 22. Change the corresponding numbers according to the main manuscript.**

The line has been modified to the following:

*The results show that the change in concentration indicated by the SOAmod scheme is stronger especially for summertime, showing a difference of +113%, +156% and +263% for SOA2p, SOAvbs and SOAmod respectively, for the European area.*

**24. P16 line 4. Change the corresponding numbers according to the main manuscript. Currently are not consistent.**

The line has been changed to the following:

*the changes for this region are stronger in the SOAmod scheme as well (76\%, 75\% and 127\% for SOA2p, SOAvbs and SOAmod respectively for summer).*

**Referee 2 minor comments:**
    **1. P2 line 12. Remove commas after the dots in reference.**

Modified.

    **2. P2 line 31. Remove comma after the dot in reference.**

Modified.

    **3. P2 line 32. Define CHIMERE.**

As far as I know, CHIMERE does not have a definition, it's a specific name given to the model.

    **4. P2 line 34. Remove all the dots in µg.m-3**

Modified.

    **5. P2 line 35. Remove comma after the dot in reference.**

Modified.

6. **P5 line 12. Remove commas after the dots in reference.**

Modified.

7. **P5 line 31. Add a dot after Cholakian.**

Modified.

8. **P7 line 12. Define ECMWF.**

Modified.

9. **P7 line 13. Define MEGAN.**

Modified.

10. **P8 lines 11-12. Remove dots in molecules.cm2**

Modified.

11. **P8 Figure 3. Check the order of the months. June-July-August are before March-April-May. Please magnify the table. Replace bials with bias.**

Modified.

12. **P9 Figure 4. Line2. Please add next to historic simulations, also future simulations.**

Modified.

13. **P10 line 12. Change august to August.**

Modified.

14. **P12 Figure 6. Remove dots from units.**

Modified.

15. **P15 line 6. Change VBS mod to SOAmod.**

Modified.

**In this work, Cholakian et al. used three different organic aerosol simulation schemes in order to identify how they impact the calculated OA load on future climate projections. They found significant differences on the calculated biogenic SOA projections over Europe between the three OA schemes; highlighting the uncertainties that still exist on OA calculations. This study is of definite interest to the organic aerosol modeling community by contributing towards the understanding of the source of uncertainty between OA schemes (e.g., highlighting the role of temperature sensitivity). Overall, the manuscript is very well written and the presentation is clear. Therefore, I recommend this study for publication. Below are a few minor comments to be considered prior to publication.**

We would like to thank referee #3 for their pertinent remarks for this paper, we have answered these comments point by point in the section that follows.

**Referee 3 specific comments:**

1. **Title: I believe the manuscript focuses a lot on biogenic SOA, therefore is better to replace the general term "particulate matter" with biogenic SOA. Furthermore, the manuscript presents the sensitivity of BSOA concentrations on the OA scheme used and not vice versa as the title implies. I suggest to consider revising the title.**

Yes, that is true. The title of the article has been changed to:

*Biogenic SOA sensitivity to organic aerosol simulation schemes in climate projections*

2. **Page 1 lines 7-8: the word "formation" is unnecessarily repeated two times in the sentence**

Yes, the second formation has been omitted from the sentence.

3. **Page 1 lines 7-8: This sentence is not clear. I assume you men the temperature differences.**

I do not understand which sentence is referred here, probably the wrong line/page number is given in the comment?

4. **Page 2 line 1: Tsimpidi et al. (2017, doi: 10.5194/acp-17-7345-2017) is also a nice recent study that emphasizes the large uncertainty of OA formation.**

Reference added.

5. **Page 2 line 3: Lelieveld et al. (2015, doi:10.1038/nature15371) also highlight the adverse effects of OA on human health due to their increased toxicity.**

Reference added.

6. **Page 2 lines 22-25: The scheme of Pankow (1994, doi: 10.1016/1352-2310(94)90093-0) should be included in the discussion here**

Pankow (1994) is not an actual OA simulation scheme, but an OA gas-particle partitioning scheme which is also used in the CHIMERE model in order to distribute the OA into gaseous/particulate phases. It would be also appropriate to cite Odum (1996) in this part (the basis of the 2-product scheme) which has more bearing to what is being discussed. Both references have been added to this section:

*Odum (1997) suggested a two-product scheme, where he calculated yields of production of OA from VOCs from laboratory data. He concluded that two virtual semivolatile organic*

*compounds were sufficient to represent the formation of OA. Following the partitioning theory of Pankow (Pankow, 1994), these species are distributed between the aerosol and gas phases. Pun and Seigneur, (2007) suggested a molecular single-step oxidation scheme for the formation of SOA, based on the Odum scheme.*

7. **Page 4 1st pargraph: More information is needed for the simulations conducted by the global models and WRF (e.g., which RCPs were used, which is the suimulation period, etc. ?) Furthermore, are all the links between the models offline?**

The following information have been added to this paragraph:

*The WRF simulations were prepared for the EURO-CORDEX project (Jacob et al., 2014) and use representative concentration pathways (RCPs, Meinshausen et al., 2011 ; van Vuuren et al., 2011) for future simulations. The EURO-CORDEX climatic runs were performed for the period of 1976—2005 for historic simulations and 2031—2100 for future scenarios, for RCP2.6, RCP4.5 and RCP8.5. A detailed analysis of these runs is provided in Vautard et al. (2014) and Jacob et al. (2014). In this work, the RCP8.5 runs are used for a selection of years (section 2.3). Anthropogenic emissions (base year 2010) are taken from the ECLIPSEv4a inventory (Amann et al., 2013; Klimont et al., 2013; Klimont et al., 2017), and the biogenic emissions calculated with by the MEGAN model (Guenther et al., 2006). The coupling of all these models with the CHIMERE model is done in an offline fashion, except for MEGAN which is directly coupled with CHIMERE.*

8. **Page 5 lines 18-21: It has been also shown that the aging of BSOA does not lead to any net changes on its mass concentration due to a balancing effect between fragmentation and functionalization (Murphy et al., 2012, doi: 10.5194/acp-12-10797-2012)**

As far as we've seen, that is the case when BSOA aging is considered with both functionalization and fragmentation processes active. If the fragmentation processes are inactive in the scheme, the aging causes a large amount of overestimation in the concentration of BSOA. This is mentioned in Murphy et al. (2012): "The detailed functionalization case overpredicted OA concentrations at all sites and underpredicted O:C ratios considerably" (taken from Murphy et al, 2012, conclusions) and Lane et al. (2008): "Including the aging reactions for SOA compounds causes a significant increase in the total predicted SOA concentrations". Also, we have seen the same results (as in an overestimation of BSOA when only aging and functionalization are considered) in our previous study (Cholakian et al. 2018).

9. **Page 5 lines 22: Actually, the standard VBS scheme assumes that fragmentation and functionalization processes result in a net average decrease in volatility for SOA. Therefore, even if it does not simulate explicitly the fragmentation process, it has taken into account its effects on the SOA volatility changes.**

The sentence is modified to say that fragmentation is at least not considered explicitly.

*Since the standard VBS scheme does not include fragmentation processes explicitly (when molecules break into smaller and more volatile molecules in the atmosphere)…*

10. **Page 5 lines 27: Can you briefly discuss the main differences between the standard and the modified VBS schemes (e.g., aging rate constants, changes on volatility and oxygen atoms added after each reaction step, etc.?)**

Reaction rates for the common species (and generations) do not change between schemes, reaction rates for new species and generations is taken from Shrivastava et al. (2013). The aging processes are all turned on in the modified VBS scheme, two more oxidation generations are added to POAs. BSOA oxidation generations are kept the same (one generation of oxidation). The formation of non-volatile SOA is added to all the SOA oxidized species (excluding POA), forming a nonvolatile SOA which cannot return to the gaseous phase. The same fragmentation fractions reported by Shrivastava et al. (2015) are used without any change. This same explanation has been added to the text at the specified part as well. The following paragraph has been added to the article:

> *Reaction rates for the common species (and generations) do not change between these two schemes, reaction rates for new species and generations is taken from Shrivastava et al. (2013). The aging processes are all turned on in the modified VBS scheme, two more oxidation generations are added to POAs. BSOA oxidation generations are kept the same (one generation of oxidation). The formation of non-volatile SOA is added to all the SOA oxidized species (excluding POA), forming a nonvolatile SOA which cannot return to the gaseous phase. The same fragmentation fractions reported by Shrivastava et al. (2015) are used without any change.*

**11. Page 7 lines 22: Can you comment on why all schemes significantly fail to reproduce the observed OA concentrations during winter?**

It has been documented that the VBS and the molecular scheme both tend to underestimate the concentration of formed OA in urban areas (Bergström et al., 2012, Ciarelli et al., 2016). This is mostly because of sources lacking in the emissions inventories, especially for the residential sector (biomass burning). Since wintertime OA formation comes to a high degree from this source (Louvaris et al., 2017; Roig Rodelas et al., 2019), it is logical for the wintertime OA to be underestimated. However, the goal of the paper is changes of BSOA in future scenarios and this underestimation does not affect the results of our study.

**12. Page 8 lines 11: Add "the" before "average". Furthermore, the emission units do not seem correct. You need "amount time-1 area-1". It would be better to report the emissions in Tg/yr for the whole domain.**

The modifications have been made and emissions unit has been fixed. However, since the output unit in the model for these emissions is in the unit mentioned in the article (molecules $cm^{-2}\,s^{-1}$), we would prefer to keep it the way it is.

**13. Page 9, Figure 4: The figure caption states that these are BSOA but the figure legend has SOA (i.e., SOA2p, SOAvbs, SOAmod)**

Yes, the figures are for BSOA. The SOA2p, SOAvbs and SOAmod are the names given to each scheme (as explained in section 2.2) and do not mean that the SOA is shown.

**14. Page 9 lines 6-9: I found this sentence long and confusing**

Yes, the sentence seems to be too long and confusing and has been modified:

> *These changes show that the climate impact on changes of BSOA in the future might be underestimated until now on a relative scale. This is because most of the future simulations performed in order to explore climate impact use a two-product or a molecular single step scheme for the simulation of SOA. However, our study shows that using a VBS based scheme increases the climate induced effect on the change in BSOA concentration in the future.*

**15.Page 10 lines 12: use capital A for August**

Modified.